# Assessment of food trade impacts on water, food, and land security in the MENA region

Sang-Hyun Lee[1], Rabi H. Mohtar[2], Seung-Hwan Yoo [3]

[1] Research Institute for Humanity and Nature (RIHN), Motoyama 457-4, Kamigamo, Kita-ku, Kyoto 603-8047, Japan

[2] Department of Biological and Agricultural Engineering, Texas A&M University, College Station, USA and Faculty of Agricultural and Food Sciences, American University of Beirut, Beirut, Lebanon

[3] Department of Rural and Bio-systems Engineering, Chonnam National University, Gwangju, Republic of Korea

*Correspondence to*: Rabi H. Mohtar (mohtar@tamu.edu and mohtar@aub.edu.lb)

**Abstract**

The Middle East and North Africa (MENA) region has the largest water deficit in the world. It also has the least food self-sufficiency. Increasing food imports and decreasing domestic food production can contribute to water savings and hence to increased water security. However, increased domestic food production is a better way to achieve food security, even if irrigation demands increase in accordance to projected climate changes. Accordingly, the trade-off between food security and the savings of water and land through food trade is considered as a significant factor for resource management, especially in the MENA. Therefore, the aim of this study is to analyze the impact of food trade on food security and water-land savings in the MENA region. We concluded that the MENA region saved significant amounts of national water and land based on the import of four major crops, namely, barley, maize, rice, and wheat, within the period from 2000 to 2012, even if the food self-sufficiency is still at a low level. For example, Egypt imported 8.3 million ton/year of wheat that led to 7.5 billion m³ of irrigation water and 1.3 million ha of land savings. In addition, we estimated the virtual water trade (VWT) that refers to the trade of water embedded in food products and analyzed the structure of VWT in the MENA region using degree and eigenvector centralities. The study revealed that the MENA region focused more on increasing the volume of virtual water imported during the period 2006–2012, yet little attention was paid to the expansion of connections with country exporters based on the VWT network analysis.

Keyword: *Food security; Food self-sufficiency; Food trade; Virtual water; MEAN.*

## 1 Introduction

Food security and water scarcity are urgent socio-economic and environmental issues in the Middle East and North Africa (MENA) region (Saladini et al., 2018), which are highly interlinked, and Water-Energy-Food Nexus has been suggested as a proper and integrated approach for resource management (Bazilian et al., 2011; Rasul, 2014; Mohtar and Daher, 2014; Lee et al., 2018). For example, food security in the MENA region has become complicated by increased risks owing to the geopolitical challenges and inability to satisfy needs with domestic production because of the lack of adequate arable land and water resources (Rastoin and Cheriet, 2010). In addition, food imbalance in the MENA region is forecast to reach 60 % in 2050 and food security in MENA region could be extremely compromised (Rastoin and Cheriet, 2010). Climate change could lead to more frequent occurrence of extreme climatic events in Mediterranean region, accompanying 50 % decrease of agricultural production by the end of the century (Porter et al., 2014). In particular, water saving through food trade can be suggested as a solution for mitigating groundwater depletion in the MENA region (Lezzaik et al., 2018).

In this study, we focused on the role of food trade in the MENA region in terms of resource management. Accordingly, we applied the concept of virtual water trade (VWT), which refers to the trade of water embedded in food products (Allan, 1993; Aldaya et al., 2010; Antonelli and Tamea, 2015), in order to assess the food trade impact on water savings in MENA region. International trade in food commodities has been shown to save water, thus food trade is an important element of both food

and water security in water-scarce regions (Hoekstra, 2003; Chapagain et al., 2006; Hanjra and Qureshi, 2010; Fader et al., 2011; Konar et al., 2012). In addition, food trade could contribute to global water savings if food is exported by countries with a higher water productivity than the countries of import (Konar et al., 2012). The concept and quantitative estimates of virtual water can help to realistically assess water scarcity for each country, projecting future water demand for food supply, thus increasing public awareness on water and identifying water-wasting processes in production (Oki and Kanae, 2004). For water-scarce countries, achieving water security by importing water intensive products could be a more attractive option compared to producing all water-demanding products domestically (Hoekstra and Hung, 2005). The global volume of international crop-related virtual water flows averaged 695 billion m³/year over the period 1995–1999, which means that 13% of the water used for crop production in the world was not used for domestic consumption but rather for export in virtual forms (Hoekstra and Hung, 2005). Falkenmark and Lannerstad (2010) estimated that it would be necessary to double the VWT by 2050 to compensate for agricultural water deficits because of climatic change, population increase, and the pattern of food supply per capita. For example, an average of 20% of the per capita food energy supply was assumed to originate from animal foods to ensure sufficient protein content, and additional water was required to produce animal foods compared to other food types (Falkenmark and Lannerstad, 2010).

The VWT could contribute to the relief of water stress through the use of global water in a more efficient manner in the event of an increase in the global food trade (Molden, 2007). Additionally, the VWT and the respective savings garnered through the trade of agricultural goods have been quantified in a number of studies. Oki and Kanae (2004) investigated that approximately 1140 km³/year of virtual water could be used for altering the import of food products to domestic products, e.g., cereals, soybeans, and meat; however, 680 km³/year of water was used to produce these food types in exporting areas. Yang et al. (2006) revealed that the VWT could generate global water savings because virtual water has flown primarily from countries of increased crop water productivity to countries of low-crop water productivity. In their study, 336.8 km³/year of water were saved globally by the international trade of major food crops from 1997 to 2001, while 20.4% of the total global net virtual water import was imported by countries that have water availability below 1700 m³ per capita, such as the Arab countries. Fader et al. (2011) calculated the VWT based on the trade of crop products, and compared it with the water requirements for producing crop products in each country for domestic consumption without international trade. Generally, exporters use less water for production of crop products than importers. Thus, the trade of crop products saves 263 km³/year of water globally, thereby representing 3.5% of the annual precipitation on cropland (Fader et al., 2011). In particular, water-scarce countries, such as China and Mexico, as well as land-scarce countries such as Netherlands and Japan, saved large amounts of water by importing goods that require water in the range from 25 to 73 km³/year, because they would otherwise need relatively large amounts of water to produce the goods they import. According to the study by Biewald et al. (2014), blue water, which refers to the irrigation water supplied from artificial facilities, such as reservoirs, ground water pumping or desalination stations, was saved in importing countries by importing products in accordance to international trade. It is expected that this can elicit enormous benefits in water-scarce regions. For example, 17 billion m³ of blue water per year were saved by the global food trade, and the value of blue water saving was estimated to 2.4 billion US$.

Previous studies showed that the effective import of virtual water may reduce water use for domestic food production in importing countries and help alleviate water stress in the MENA region where the largest water deficit in the world exists (Gleick, 2000; World Bank, 2009). The critical condition of water scarcity in the MENA region will reach severe levels by 2025 (Tolba, 2009). In addition, if population increases rapidly and urbanization continues fast, availability of water could be reduced in the Arab countries by approximately 50% by the year 2025 (Abahussain et al., 2002). Water shortages will certainly speed up the rate of desertification in the Arab countries (Abahussain et al., 2002). Agricultural water withdrawals account for over 85% of the total water withdrawn by the various countries of the MENA region (FAO, 2014). Irrigation systems in the MENA region are based on pumping groundwater resources, such as aquifers, and water security is being threatened by the declining aquifer levels and the extraction of nonrenewable groundwater (Antonelli and Tamea, 2015). In addition, Immerzeel

et al. (2011) expected that the unfulfilled water demand in the entire MENA region would increase from the current level of
16% to 51% in 2040–2050 owing to climate changes. The zone of severely reduced rainfall extends throughout the
Mediterranean region and the Northern Sahara (Hennessy et al., 2007). Milly et al. (2005) estimated that climate change will
cause a decrease in water run-off by 20% to 30% in most of the MENA region by 2050, mainly owing to the rising temperatures
and lower precipitation. In addition, the regions that include Syria, Lebanon, Israel, and Jordan, will get drier, with significant
rainfall decreases in the wet season.
However, the high dependency on food import can be a risk of food security, even if it can elicit domestic water, energy, and
land savings, in water-scarce regions. Therefore, we should consider a trade-off between food security and resource savings,
using a holistic approach, such as Trade-WFL(Water-Food-Land) Nexus. Furthermore, the VWT can be suggested as relevant
to the water policy of a nation (Schyns and Hoekstra, 2014), thus establishing a new point-of-view from which both food
security and sustainable water management are considered (Novo et al., 2009).
This study addresses three questions that relate to the role and impact of the VWT in the MENA region, that are raised to draw
attention to the complexity of the issue and the need for a broader view in assessment. Specifically, 1) what are the effects of
the VWT on water savings and land tenure in the MENA region, 2) has the structure of the virtual water import in the MENA
region been vulnerable or robust? 3) Who are the influential importers and exporters in the VWT network in the MENA region?
The aim of this study is to evaluate the effects on water savings and land tenure from importing crops  at 15 countries in the
MENA region such as Algeria, Egypt, Iraq, Jordan, Kuwait, Lebanon, Libya, Morocco, Oman, Qatar, Saudi Arabia, Syria,
Tunisia, UAE, and Yemen. In addition, we quantified the amount of VWT from 2000 to 2012, and analyzed a structure of the
VWT, such as the connectivity and influence in the MENA region using degree and eigenvector centralities.
**2 Materials and Methods**
**2.1 VWT based on international trade**
The VWT represents the water embedded in international trade, and it indicates the water used in the exporting country to
produce crops for export. Therefore, the VWT is calculated based on the water footprint of exporters, which indicates the total
amount of water used for producing crop, and the export of virtual water in the exporting country has the same meaning as the
import of virtual water has in the importing country. For example, Saudi Arabia imported wheat from various exporters, and
the virtual water import(or export) was calculated by multiplying the quantity of traded wheat with the respective water
footprint of exporters. Accordingly, the main factors for quantifying a VWT are the trade data and water footprint, and the
VWT is calculated by multiplying the trade by its associated water footprint in the exporting country, as follows:
$VWT\,[n_e, n_i, c, t] = CT\,[n_e, n_i, c, t] \times WFP\,[n_e, c],$         (1)
where the variable VWT denotes the virtual water trade from the exporting country, $n_e$, to the importing country, $n_i$, in year
t, as a result of trade in crop c, CT represents the crop trade from the exporting country, $n_e$, to the importing country, $n_i$, in
year t as a result of trade in crop c, and WFP represents the water footprint of crop c in the exporting country, $n_e$.
The international trade data of the four major crops, namely, barley, maize, rice, and wheat from 2000 to 2012 was obtained
from FAOSTAT (http://www.fao.org/faostat/), as shown in Table 1. The crop with the largest amount of import was wheat,
with 27.6 million ton/year imported by the MENA region from 2000 to 2012, followed by maize (14.4 million ton/year), barley
(9.0 million ton/year), and rice (3.7 million ton/year).
Water footprint is a localized index for countries, accounting for the climate, productivity, and irrigation. In this study, we
considered water footprints of all countries in the world, however, a lot of effort should be required for estimating water
footprints of all countries and it was outside the scope of the current study. Therefore, we applied water footprint data of 147
countries, including those in the MENA region, from the study executed by Mekonnen and Hoekstra (2010). The water
footprint for a crop is divided into green and blue water footprints based on the water resources (Hoekstra and Chapagain,
2008). The green water footprint indicates that water supplied by precipitation is retained in the soil of the root zone
(Falkenmark, 1995), and blue water footprint is the water stored at the surface or in the ground. Therefore, the green water
footprint is related to rain-fed agriculture and the blue water footprint is related to irrigation water provided by aquifers or
surface bodies of water. As the water footprint is divided into green and blue water footprints, water saving could be considered
as green and blue water saving as well.
**Table 1.** Cultivation area, production, the quantity of crops imported, and internal water resource in the MENA region from
2000 to 2012

**2.2 Water and lands savings by an international food trade in importing country**

Food import is also related to domestic water and lands savings. In particular water saving has a different meaning from virtual
water import. For example, Saudi Arabia imported wheat from various exporters and virtual water import indicates the sum of
the products obtained from multiplying the quantity of imported wheat by the respective water footprint of each exporter.
However, water saving indicates the amount of water needed to produce the same quantity of imported products domestically.
Therefore, water saving by wheat import in Saudi Arabia is estimated by multiplying the quantity of imported wheat with the
water footprint of wheat in Saudi Arabia.
In this study, we applied green and blue water footprints of crops in each country in the MENA region, as shown in Table 1.
However, the availability of water footprint data in the MENA region was limited in some cases. For example, the water
footprint of wheat was available in all countries except for Bahrain. Lands saving has the same implication as water savings,
thus we calculated lands saving using land footprint of each country in the MENA region, as shown in Table 2. The land
footprint indicates the land requirement for producing 1 ton of crops, and it was calculated based on the harvest area and crop
production data collected from FAOSTAT (Table 1).
The water and lands savings could be assessed the impacts of failure of trade on domestic water and land requirements in the
importing country. Although this assumption about water and land savings considers an extreme trade situation, these results
could be used to understand the importance of the international crop trade in the MENA region. In other words, the water and
land savings indicated the amount of water and land requirements for crops imported to substitute domestic production, and
the water and land savings were calculated as follows,
$WFP\,[n_i, c] = \frac{CWR\,[n_i, c]}{P\,[n_i, c]}$                              (2)
$LFP[n_i, c] = \frac{Area\,[n_i, c]}{P\,[n_i, c]}$                              (3)
$WS\,[n_i, c] = CI\,[n_i, c] \times WFP\,[n_i, c],$                       (4)
$LS\,[n_i, c] = CI\,[n_i, c] \times LWP\,[n_i, c]$                       (5)
in which variable WFP $[n_i, c]$ (m³/ton) is the water footprint of crop c in the importing country $n_i$, CWR is the crop water
requirement (m³), and $P$ is the production (ton). Equivalently, LFP$[n_i, c]$ (ha/ton) is the land footprint of crop c in the importing
country $n_i$, and $Area$ is the cultivated area (ha). The symbol WS (m³) or LS (ha) indicates the amount of water or land savings
in the importing country $n_i$. CI is the import of crop c in the importing country $n_i$.
**Table 2.** Water and lands footprints of four major crops in the MENA region

**2.3 Degree and eigenvector centralities for analyzing the structure of VWT**

*2.3.1 Nonscaled and scaled in-degree centralities of VWT*
Understanding the VWT structure is important for quantifying the amount of import and export because the VWT structure
can represent whether it would be sustainable or vulnerable. For example, if a country imports considerable amounts of virtual
water through the food trade from just a few exporters, the structure of VWT in this country might be impressionable by
exporters. However, if a country is connected with many exporters in VWT, it can have a resilient structure for global changes.
A few studies have been conducted on the analysis of the structure of the VWT using a network-based approach (Konar et al.,
2012; Dalin et al., 2012; Lee et al., 2016). For example, Konar et al (2012) analyzed the characteristics of the network change
in virtual water trade (VWT), and found that a number of export trade partners followed an exponential distribution in 2000.
Dalin et al (2012) found that constant organizational features were observed in the network of VWT even though the number
of trade connections and the volume of VWT has been growing. In addition, Lee et al (2016) analyzed vulnerability of the
importing countries through the characteristics of network in VWT.
In this study, we analyzed the links of the VWT network for identifying the VWT structure using degree centrality, that is the
number of degree incidents on a given node (Freeman 1979). In addition, the degree centrality is divided into in- and out-
degree centralities, depending on the direction. In-degree is based on the number of lines (or volume) directed to the node. and
out-degree is based on the number of lines (or volume) that the node directs to. A node indicates the country in global trade
network, and incidents mean the trade between countries which can be amounts of products or number of connections, fox
example if one country exports product to five countries, that country has five incidents. In this study, we focused on the in-
degree centrality because the MENA region includes representative importing countries. An importer accompanying an
increased in-degree centrality has expanded connectivity with a large number of exporters, meaning that this importer could
cope with an accidental disconnection from a certain exporter. In addition, the volume of products exported or imported can
be applied to incidents as weight of links. In this study, the in-degree centrality, based on the VWT network, is expressed
according to the nonscaled in-degree centrality (NSInDC), that is based on the number of links, and the scaled in-degree
centrality (SInDC), that is based on the volume of links.
$NSInDC_i = \sum_j^N Link_{ij}/(N-1),$ (6)
$SInDC_i = \sum_j^N Flow_{ij}/(N-1),$ (7)
where $NSInDC_i$ is the nonscaled in-degree centrality of country i, and $Link_{ij}$ is the number of links between the ith and jth
countries. The symbol $SInDC_i$ is the scaled in-degree centrality of country i, and $Flow_{ij}$ is the volume of virtual water traded
between the ith and jth countries. Moreover, N is the total number of countries that trade with a given MENA countries.
Through NSInDC and SInDC, we analyzed the vulnerable expansion (or reduction) and robust expansion (or reduction) in the
VWT network in the MENA region. For example, the vulnerable expansion in the network indicates that the amount of flow
to a node increases but the number of connections to other nodes decrease. This is represented by high levels of SInDC and
low levels of NSInDC. The importer country that is associated with vulnerable expansion has an increased quantity of products
from only a few exporters.

*2.3.2 Eigenvector centralities of VWT*

In general, connections to nodes which are themselves influential could make a node more influence than connections to less
influential nodes (Newman, 2016), and eigenvector centrality can be used for measuring the influential connections (Ruhnau,
2000). For example, the concept of eigenvector centrality has been used by the Web search engine Google in order to rank
Web pages (Berry and Browne, 2005; Bryan and Leise, 2006; Newman, 2016).
In VWT network, the eigenvector centrality could be used for identifying influential countries that could affect the entire
network. In other words, the entire VWT can be affected by a few influential countries, and it is important to identify these
countries for understanding and estimating the change of the entire structure of the VWT. An eigenvector centrality can
measure the influence of each country in the entire VWT, and it is related not only to its own connection pattern but also to
the connections of other countries to it. Therefore, a country is more influential if it is considered in relation to the countries
that are influential themselves (Ruhnau, 2000). The eigenvector centrality assigns relative centrality to all of the countries in
the VWT, based on the principle that connections to high-level centrality countries contribute more to the centrality of the
countries compared to equal connections to low-level centrality countries (Ruhnau, 2000; Lee et al., 2016). Bonacich (1972)
defined the centrality $(x_i)$ of a node i as the positive multiple of the sum of adjacent centralities in links (or volume) between
nodes ($A_{ij}$). Therefore, if we denote the centrality of vertex i by $x_i$, then we can allow for this effect by making $x_i$ proportional
to the average of the centralities of i's network neighbours (Newman, 2016),
$x_i = \frac{1}{\lambda}\sum_{j=1}^{n} A_{ij}x_j$ (8)
where λ is a constant. Defining the vector of centralities x = ($x_1$, $x_2$,...), we can rewrite this equation in matrix form as
λx = Ax (9)
This type of equation is solved using eigenvalues and eigenvectors, where A is a adjacency matrix of $A_{ij}$, and λ is a scalar,
known as the eigenvalue associated with the eigenvector c defined as a column vector. Eigenvector centrality is determined
by calculating the principal eigenvector that has the largest eigenvalue among all eigenvectors. A non-negative eigenvector
with the maximal eigenvalue exists. We refer to a non-negative eigenvector (x ≥ 0) of the maximal eigenvalue as the principal
eigenvector, and we call the entry $x_i$ the eigenvector-centrality of node (country) i (Ruhnau, 2000).
**3 Results and Discussion**
**3.1 Trade-offs between national water-land savings and food security through food trade in the MENA region**
This study considered trade-offs between food security and food trade in terms of national resource management. For example,
the increase of domestic food products instead of imports of them could be one policy for food security but additional water
and land for domestic products would be considered at the same time. In other words, food imports could contribute domestic
water and land management, therefore, we estimated the national water and land savings by importing crops as shown in Table
3. In Saudi Arabia, blue water savings by barley, maize, and wheat imports were estimated to 5.0, 2.0 and 0.8billion m³/year,
respectively. In comparison to the internal water resource of Saudi Arabia which is 2.4 billion m³/year as shown Table 1. the
water saving through import of barley, maize, and wheat could be considered as significant amount in Saudi Arabia. In the
case of Egypt, most of the water saving occurred based on the imports of wheat and maize. Approximately 7.5 billion m³/year
of blue water was saved by importing wheat. Specifically, the internal water resources in Egypt are only 1.8 billion m³/year
(Table 1), therefore, water scarcity could be an issue for food security policy in Egypt. Lebanon was strongly influenced by
the impact of crop import on land savings. Approximately 0.24 million ha could be saved by crop imports, comprising 36% of
the agricultural area in Lebanon, that indicates that the crop trade in Lebanon has significant benefits in terms of land resources
compared to water resources.
Food imports could be regarded as a negative factor in food security, and it is obvious that food security would accompany
water and lands for domestic food products. These results showed that food imports could bring positive impacts on numerous
water and lands savings in the MENA region. However, there are limitations of these results. First, water saving estimated in
this study was based on the hypothetical situation that meat there were no international trade situation, and sometimes it was
larger than the internal water resources in some countries such as Saudi Arabia and Egypt. Additionally, some crops are
required for the specific type of climate but this study assumed that MENA region was suitable for cultivating maize, wheat,
barley, and rice.
**Table 3.** The amount of water and land savings through importing crops in the MENA region from 2000 to 2012.
**3.2 The VWT in the MENA region from 2000 to 2012**
*3.2.1 Virtual water import in the MENA region*
The total amount of green and blue water imported by each MENA country from 2000 to 2012 respectively reached 921.2 and
80.5 billion m³ in the MENA region, as shown in Table 4 and Figure 1. The largest volume of green water was imported
annually by Egypt (19.1 billion m³/year), followed by Saudi Arabia (11.9 billion m³/year). In addition, the largest amount of
blue water was imported annually by Saudi Arabia (1.2 billion m³/year), followed by the UAE (0.9 billion m³/year). Over 70%

of the green water imported annually into the MENA region based on the trade of barley (approximately 8.5 billion m³/year) was occupied by Saudi Arabia. The amount of virtual water imported based on the trade of maize was 13.0 billion m³/year, with Egypt being the primary importer of 31% of the total imported amount into the MENA region.

Generally, rice is cultivated in paddy fields, and the blue water footprint of rice in these fields is larger than other cereal crops in various countries. For example, the global average of the blue water footprint of rice is 584 m³/ton but that for wheat is 343 m³/ton (Chapagain and Hoekstra 2011; Mekonnen and Hoekstra 2010). Therefore, the importers of rice also import a lot of water. Approximately 3.0 billion m³/year of blue water were imported in the rice trade from 2000 to 2012, and Saudi Arabia, UAE, and Iraq, were the primary importers. The largest volume of virtual water imported by the MENA region was owing to the trade of wheat. The annual amount of virtual water imported based on the trade of wheat in the MENA region from 2000 to 2012 was approximately 42.6 billion m³/year, and . over 35% of the virtual water imported through the wheat trade was imported by Egypt (15.7 billion m³/year). However, the amount of blue water was only 2.0 billion m³/year because the green water footprint is much larger than blue water footprint in main exporters such Russian fed, Australia, and Canada that might indicate wheat has been cultivated in rain-fed area with less irrigation.

We also estimated the amount of virtual water imported per capita (VWIcap), as shown in Figure 2, which shows the differing viewpoints regarding food and water securities. If we consider only the total amount of imported virtual water, the UAE may not be considered to be a significant importer because the population and area of UAE is much smaller than those of the MENA other countries, such as Saudi Arabia. However, the virtual water import per capita in the UAE is larger than that of Saudi Arabia, thus indicating that the dependency on virtual water imported from exporters in the UAE is much more significant than in Saudi Arabia. For example, the VWIcap was 1266.6 m³/cap/year in the UAE, which was the largest value in the MENA region. The UAE is strongly dependent on the import of virtual water, even though the UAE imports only 4.2 billion m³/year of virtual water. The VWIcap increased significantly in Saudi Arabia and Libya from 2000 to 2012. Saudi Arabia and Libya imported approximately 453.4 and 497.8 m³/cap/year, respectively, of virtual water more in 2012 than in 2000. Saudi Arabia was the second largest importer in the MENA region, and its VWIcap was also the fifth highest in the MENA region.

**Table 4.** The amount of green and blue water imported in the MENA region from 2000 to 2012.

**Figure 1.** The total amount of virtual water imported by each country in the MENA region from 2000 to 2012, separated into green (upper) and blue (lower) water

**Figure 2.** Virtual water imported per capita in the MENA region from 2000 to 2012.

*3.2.2 Virtual water export to the MENA region*

We also focused on the volume of virtual water exported to the MENA region by each exporter from 2000 to 2012, as shown in Figure 3. Based on the trade of barley, Ukraine exported 41.1 billion m³ of green water to the MENA region that amounted to 27% of the total green water imported in the MENA region. In terms of blue water traded through barley, five exporters (Germany, Australia, the Russian Federation, Ukraine, and India) provided 78% of the total blue water imported in the MENA region based on barley. Based on the trade of maize, Argentina contributed 40% of the total amount of green water imported by the MENA region based on maize, but the blue water imported by the MENA region was primarily from the USA. Based on the trade of rice, the major virtual water exporters to the MENA region were India, Thailand, and Pakistan. In particular, 30.4 billion m³ of blue water were imported from these countries from 2000 to 2012, which comprised 78% of the blue water imported by the MENA region based on rice. Wheat was the most representative crop imported by the MENA region. The Russian Federation and the USA provided 25% (140.6 billion m³) and 21% (111.2 billion m³) of the total amount of green water imported in the MENA region based on the trade of wheat in 2000 to 2012, respectively, and the remaining 55% was divided among several exporters, including Australia, Canada, France, and Ukraine.

**Figure 3.** Quantities of green water export (GWE) and blue water export (BWE) from the primary exporters to the MENA region from 2000 to 2012

**3.3 The temporal change of VWT structure in the MENA region**

From 2000 to 2012, both the volume and connectivity of VWT was changed. For example, the virtual water imported in the MENA region slightly increased and the VWT was distributed with more exporters in 2006, as shown in Figure 4. However, the volume of virtual water imported in the MENA region was increased more than 50 % from 2006 to 2012 but the distribution of VWT seemed to consistent. In case of Lebanon, VWT in Lebanon was strongly dependent on the USA, Argentina, and Australia. However, Lebanon expended the VWT in 2006 and Russian Federation, Turkey, and Kazakhstan, contributed to virtual water imports in Lebanon, as shown in Figure 4. Accordingly, the structure of VWT in Lebanon approached a distributed network. However, the VWT in 2012 showed that it was dominated by Ukraine and Russian Federation, though Lebanon imported more virtual water in 2012 than 2006.

**Figure 4.** Virtual water imports at the MENA region and Lebanon in 2000, 2006, and 2012

These changes are more related to the structure of VWT and the MENA region should consider not only the amount of virtual water but also the structure of VWT for sustainable food security subject to the condition of a strong dependency on crop import. Therefore, we analyzed the degree centralities of NSInDC and SInDC from 2000 to 2012 in the MENA region, and identified the countries who had the vulnerable expansion or reduction in the VWT network. Figure 5 shows the NSInDC and SInDC patterns in the VWT network in accordance to each country in the MENA region. If the specific country has both large NSInDC and small SInDC, this country has connections with various exporters but imports a small amount of virtual water. Specifically, Egypt and Yemen showed that NSCInD was lower but SInDC was higher than other countries, thus indicating the intensive connectivity with a few exporters. In contrast, Saudi Arabia had larger SInDC than other countries expect for Egypt, while the NSCInD was also highest of the MENA region. Accordingly, Saudi Arabia had a more distributed structure regarding VWT. UAE and Iraq had similar SInDC in 2012 but NSInDC was quite different (UAE (0.46) and Iraq (0.27)). Furthermore, SInDC in Morocco (96.45) was larger than UAE (83.41) but NSInDC in Morocco (0.26) was smaller than UAE (0.46). In comparison to UAE, Morocco had intensive connections with fewer exporters compared to UAE.

Based on the temporal changes of NSInDC and the SInDC during two periods (2000–2006 and 2006–2012), the MENA region countries were divided into four types (I–IV), as shown in Figure 6. The x-axis indicates the NSInDC and the y-axis indicates the SInDC. Type I countries is located at higher levels both in the x-axis and y-axis, and show a robust expansion in the virtual water import. Additionally, the countries in this type increased the connectivity and volume of virtual water imported, simultaneously. Type II countries increased the volume of virtual water imported without expansion of connectivity. Type III countries showed reductions in the virtual water import with reduction of connectivity, and type IV countries has established connections with more exporters but has decreased virtual water imports.

In the early 2000s, most of countries in the MENA region expanded their trade structure by increasing both the connectivity to the exporters and the volume of the imported virtual water. In Bahrain, Omen, Qatar, Yemen, Saudi Arabia, Lebanon, and UAE, the NSInDC of the VWT network increased significantly from 2000 to 2006, which means that the trade connectivity expanded. The expanded structure of the VWT indicates that the MENA countries were connected to various exporters, and that this structure can be a resilient structure for global changes. In particular, the import of food crops is an essential factor in food security in the MENA region, even if food self-sufficiency is increased by increasing domestic production. However, Egypt had the largest SInDC but NSInDC was ranked 6th among the MENA region countries. In 2006, Egypt and Saudi Arabia both expanded the connectivity in the VWT network, as shown by the increasing NSInDC. However, the type of VWT structure in many MENA countries such as Yemen, Qatar, Bahrain, and Lebanon has moved to Type II which means that the countries increased the volume of the imported virtual water, but the number of exporters that linked to the MENA countries decreased from 2006 to 2012. In particular, in 2012, most countries kept their connectivity or reduced them, except for Algeria, Iraq, Libya, and UAE. These results indicate that the dependence of the MENA region on virtual water import increased rapidly

recently with the large increase in the imported volume of virtual water. However, the connectivity of the VWT in the MENA region has not increased as much as the volume of virtual water imported increased.

The degree centrality in this study could be useful for identifying the connectivity and volume of trade of each country, but it is limited to show the influence of each country on entire trade network, thus we estimated eigenvector centrality, as shown on Figure 7. In 2000, Egypt and Saudi Arabia were identified as the most influential importers in the MENA region, and the USA and Australia were the most influential exporters. Accordingly, the entire VWT in the MENA region could be affected by these importers and exporters. This means that the change of the trade policy or food management in these countries could change the structure of VWT in the MENA region. In 2006 and 2012, the influential countries in the MENA region were still Egypt and Saudi Arabia, but the influential exporters moved to the Russian Federation, Ukraine, and Brazil.

**Figure 5.** Nonscaled and scaled in-degree centralities of each country in the MENA region in 2000, 2006, and 2012

**Figure 6.** Country types in the MENA region according to the changes of nonscaled and scaled in-degree centralities

**Figure 7.** Eigenvector centrality of virtual water trade network in the MENA region at 2000, 2006, and 2012

**3.4 Importance and limitations of water footprint and VWT in the MENA region from a policy perspective**

Generally, the VWT is more related to resource management in exporting countries rather than importing countries because the embedded water in food trade indicates water resources that are consumed for producing food products in the exporting country. However, VWT is also considered as an important issue in importing countries in terms of water and food security. For example, the reduction of VWT might be related to water consumption by replacing imported food products by domestic food products.

However, the application of the concept of VWT is under critical discussion (Wichelns, 2010). First, water footprints formulate new concepts of water management, but we need to realize that water footprint can be changed due to various factors such water requirement, productivity, production system, development of technologies, fertilizer usage, and irrigation scheduling and operations of the water facilities. Second, VWT could contribute to the connection of water management to food security. However, food trade is affected by the scarcity or affluence of other important resources, such as capital, labor, and land (Biewald et al., 2014). In particular, economic values, such as the price of food products, are the main driver in global food trade, but there is no global value established for virtual water. Therefore, it is difficult to apply virtual water to trade policy in terms of the economic efficiency. Therefore, policy makers or resource managers in the MENA region should not only consider the effects of VWT but also the difficulty in adapting virtual water to policies for resource management. Third, there are spatial and temporal issues of VWT in the study. The VWT could be affected by geopolitical issues such as topography, and distances between importers and exporters. For example, the changes of exporting countries in the MENA region could be related to energy use for transporting products, thus trade policy should consider the economic benefit or cost of transportation. Therefore, the VWT should be discussed with geopolitical issues such as benefit and cost of transportation. In addition, VWT and water-lands savings by food trade in this study were calculated based on historical database, thus it was difficult to apply the results to future policy.

Despite these limitations, we believe that virtual water has a role in the achievement of sustainable water, land, and food security, even if there are limitations and difficulties in applying the virtual water concept. As mentioned above, the VWT can be a major resource in the MENA region. Accordingly, vulnerable VWT, for example, low connectivity, can be a risk element for future food security risk management. In particular, the MENA region is strongly dependent on food products from exporting countries which implies a strong dependency on water resource from exporting countries. Therefore, water shortages or low-food production in exporting countries might cause increasing food prices in the MENA region, but also increasing domestic water use for increasing domestic food production. The primary resources of water, energy and food are naturally interlinked. The degree of their interlinkages in the MENA is exceptionally high, thus creating a higher degree of risks and vulnerability. Therefore, understanding these interlinkages and quantifying them in an attempt to better understand this

complex system of systems is crucial. This requires the synergistic effort of multiple disciplines, including contributions from
various technologies, science, policies, health, communication, and economics, at local processes and system level scales. In
this study, we believe that the VWT in the MENA region can be the key factor for bridging water and food, and it is important
to quantify the influence of trade on water and food management. In addition, this study revealed vulnerability (or robust)
expansion (or reduction) and influential traders in the VWT network in the MENA region, based on in-degree and eigenvector
centrality indices. If a country in the MENA region has low connectivity but an increased import of virtual water, this country
should re-evaluate their vulnerable trade structure and change the trade policy or water-food management.
**4. Conclusions**
The import of water in virtual form based on VWT could develop into a major water portfolio that dominates water
management in the water-scarce countries of the MENA region. In water-deficit areas, such as the MENA region, the VWT
can offer new perspectives for understanding and solving water stress and scarcity. In summary, this study showed that the
significant water in comparison to internal water resource could be saved by food trade in the MENA region, and policy makers
can benefit by considering both the quantitative impacts of VWT and the structural changes of VWT, such as vulnerable
expansion (or reduction) in the MENA region. For example, when a country in the MENA region set a plan for increasing
food security, this country first should identify the amount of water and land savings that can be achieved by food import, and
consider the trade-off between food security and food import. In addition, the stable trade could be a component for stable
food supply in the MENA region, thus this study contributes to the understanding of the dependency on each trade partner for
countries in the MENA region and can help with setting the food trade policy in terms of extension (or reduction) of trade
partners and increase (or decrease) in volume of trade.
However, this study only focused on food trade and water-land savings, thus energy part was not considered. The MENA
region represents an extreme case globally in terms of water and energy resources, for example, 66% of the world's known
crude oil reserves, but only 1.4% of the world's fresh water supplies is attributed to the region (Khater, 2003). The increase or
decrease of water withdrawal for irrigation is related to the energy used for water extraction such as pumping surface or ground
water. For example, 5 % or more of the total electricity consumption can be attributed to water pumping in Saudi Arabia
(Siddiqi and Anadon, 2011). Energy use for food production and water supply could be the main factor in integrated resource
management in the MENA region, and the lack of energy part was a limitation in this study.
In spite of this limitation, the intensity and connectivity of VWT, which were analyzed in this study, can be the major
components needed for integrating resources management in the MENA region. Accordingly, VWT is regarded as the
important factor in determining food security and water-lands management, and it can be a useful interlinking parameter among
resources in WEF Nexus approach, which identify key issues in food, water, and energy securities through the lens of
sustainability, seeking to predict and protect against future risks and resource insecurities (Biggs et al., 2015). The core of the
Nexus concept is that the production, consumption, and distribution of water, energy, and food, are inextricably interlinked,
thus this study would provide important information to policy makers for evaluating scenarios about integrated resource
management toward sustainability in the MENA region.

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

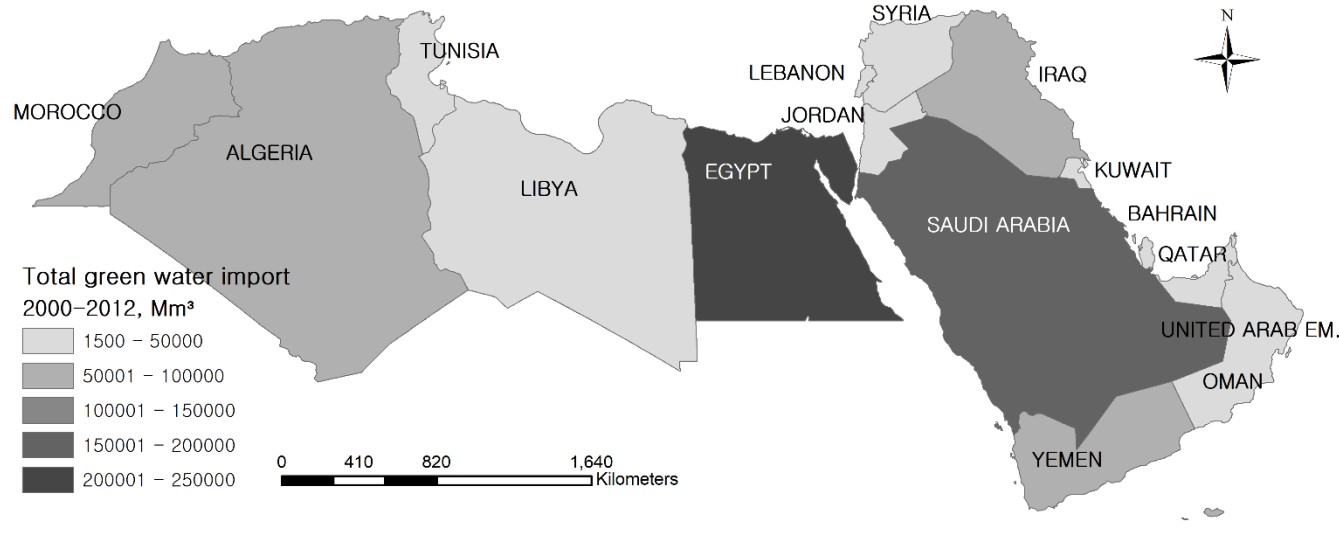

(a)  Green water imports

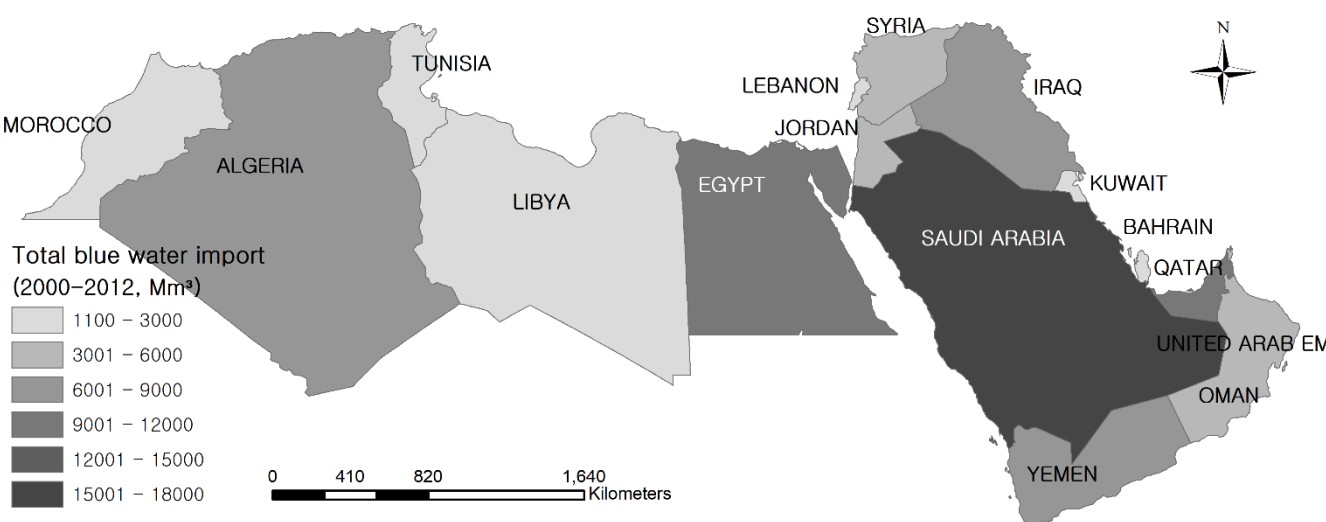

(b)  Blue water imports

**Figure 1.** Total amount of virtual water imported by each country in the MENA region from 2000 to 2012 classified into green (upper) and blue (lower) water


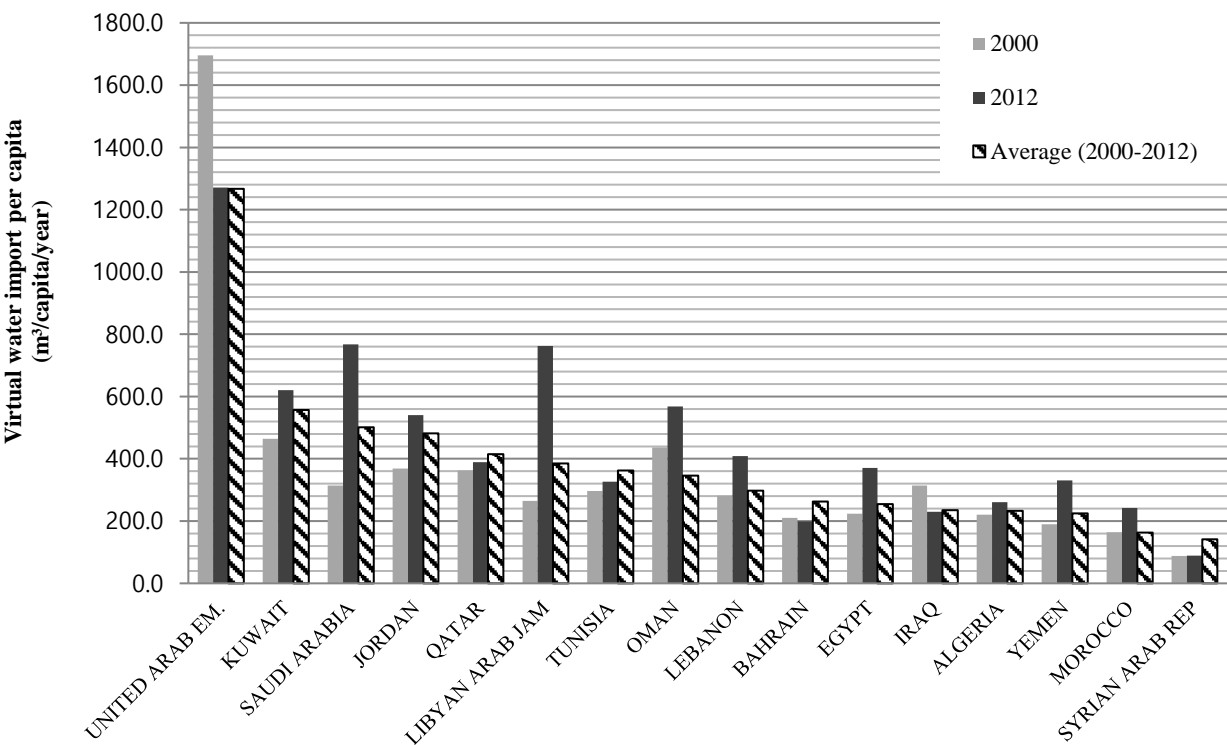


512        **Figure 2.** Virtual water imported per capita in the MENA region from 2000 to 2012


**Barley**

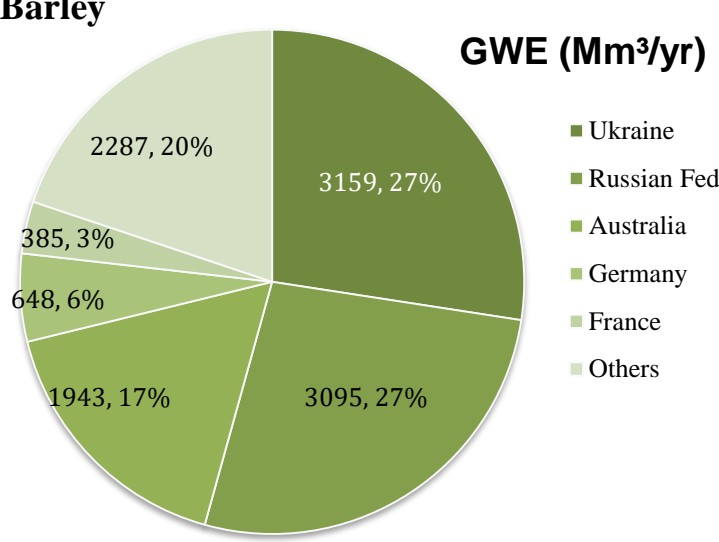

**GWE (Mm³/yr)**

- Ukraine
- Russian Fed.
- Australia
- Germany
- France
- Others


**Rice**

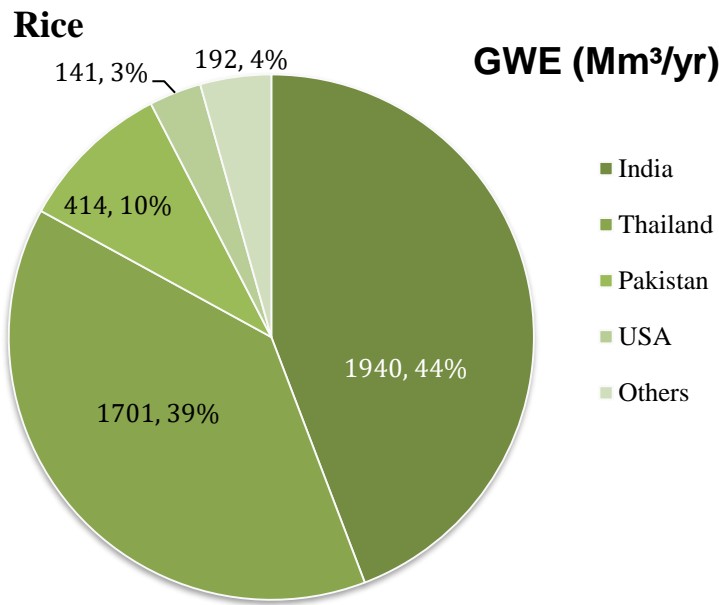

**GWE (Mm³/yr)**

- India
- Thailand
- Pakistan
- USA
- Others


**Maize**

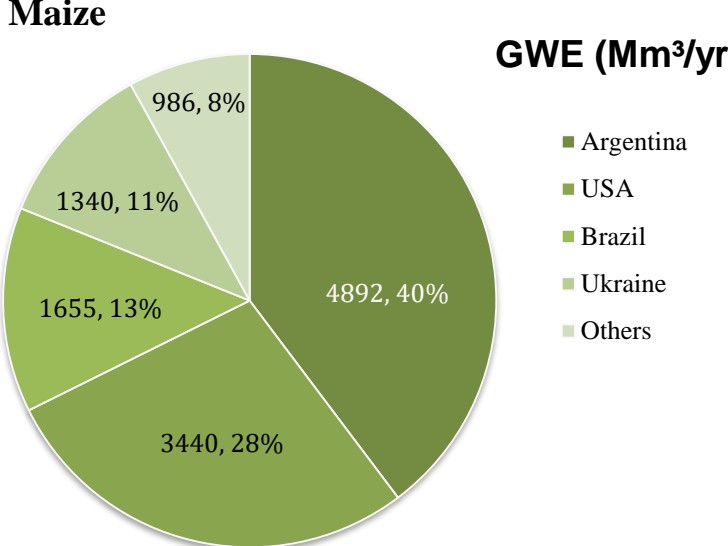

**GWE (Mm³/yr)**

- Argentina
- USA
- Brazil
- Ukraine
- Others

986, 8%
1340, 11%
1655, 13%
4892, 40%
3440, 28%


**Wheat**

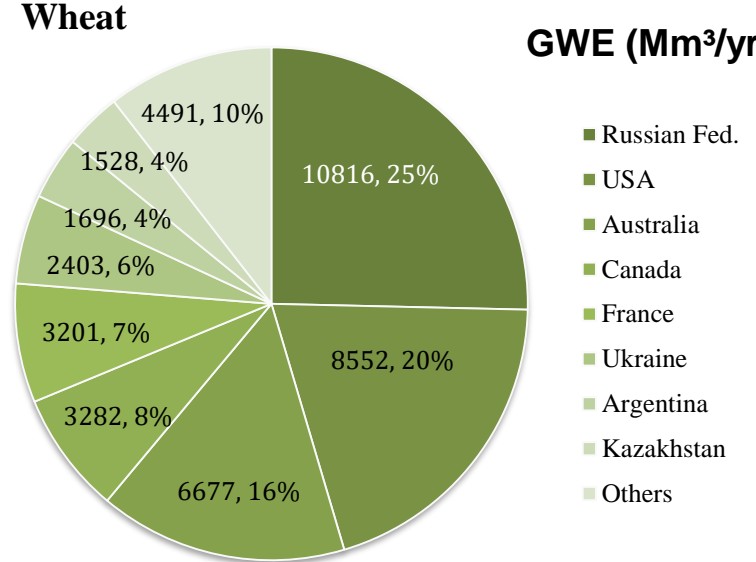

**GWE (Mm³/yr)**

- Russian Fed.
- USA
- Australia
- Canada
- France
- Ukraine
- Argentina
- Kazakhstan
- Others

4491, 10%
1528, 4%
1696, 4%
2403, 6%
3201, 7%
3282, 8%
6677, 16%
10816, 25%
8552, 20%


(a) Annaul green water export (GWE) during 2000-2012

## Barley

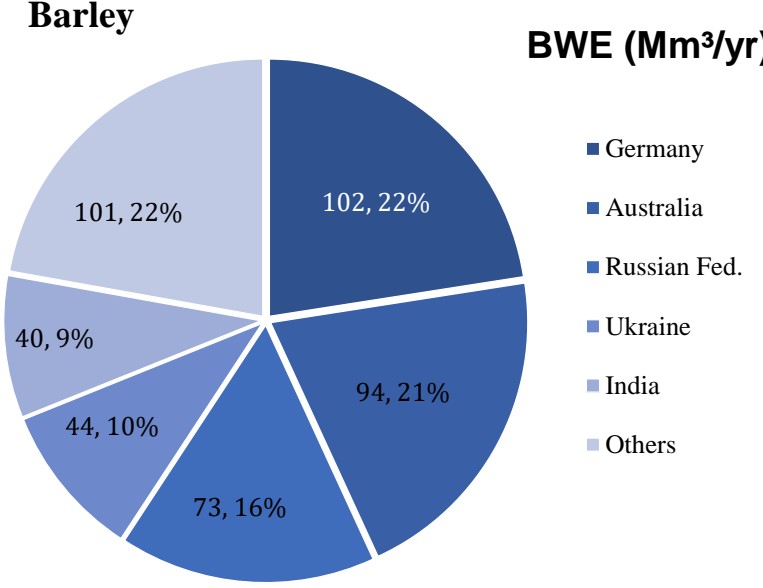

### BWE (Mm³/yr)

- Germany
- Australia
- Russian Fed.
- Ukraine
- India
- Others


## Rice

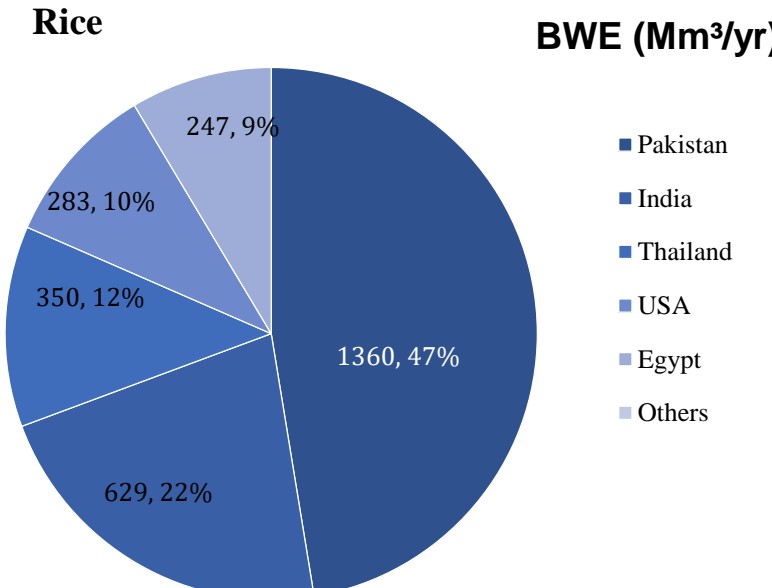

### BWE (Mm³/yr)

- Pakistan
- India
- Thailand
- USA
- Egypt
- Others


## Maize

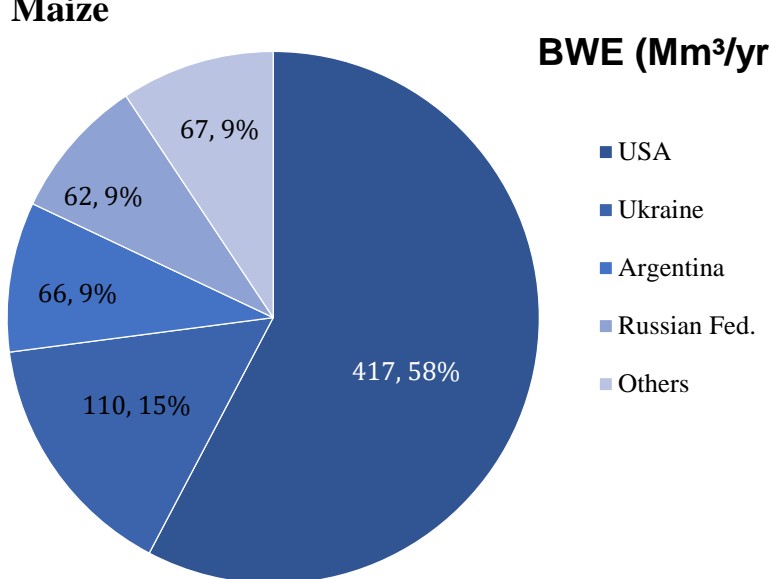

## Wheat

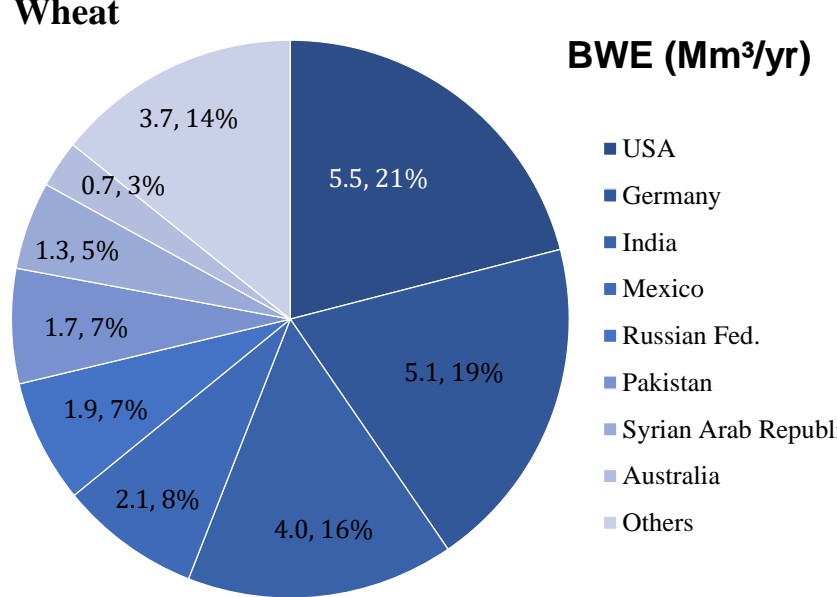


527        (b) Annaul blue water export (BWE) during 2000-2012
**Figure 3.** Quantities of annual green water exports (GWE) and blue water exports (BWE) from the primary exporters to the
530       MENA region from 2000 to 201

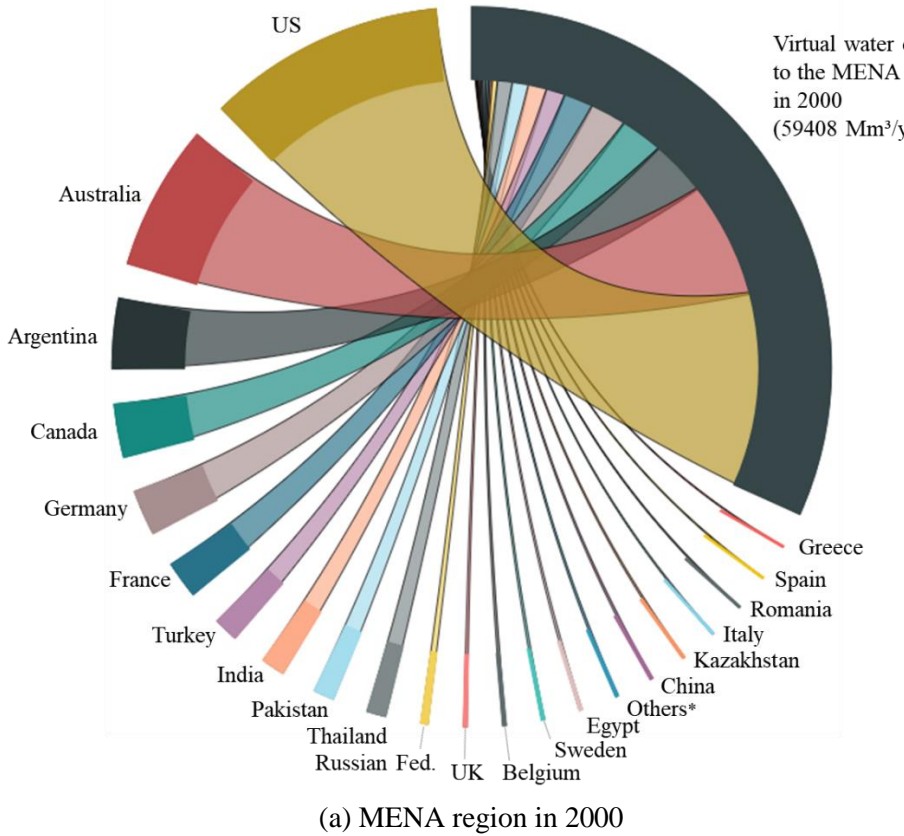

(a) MENA region in 2000

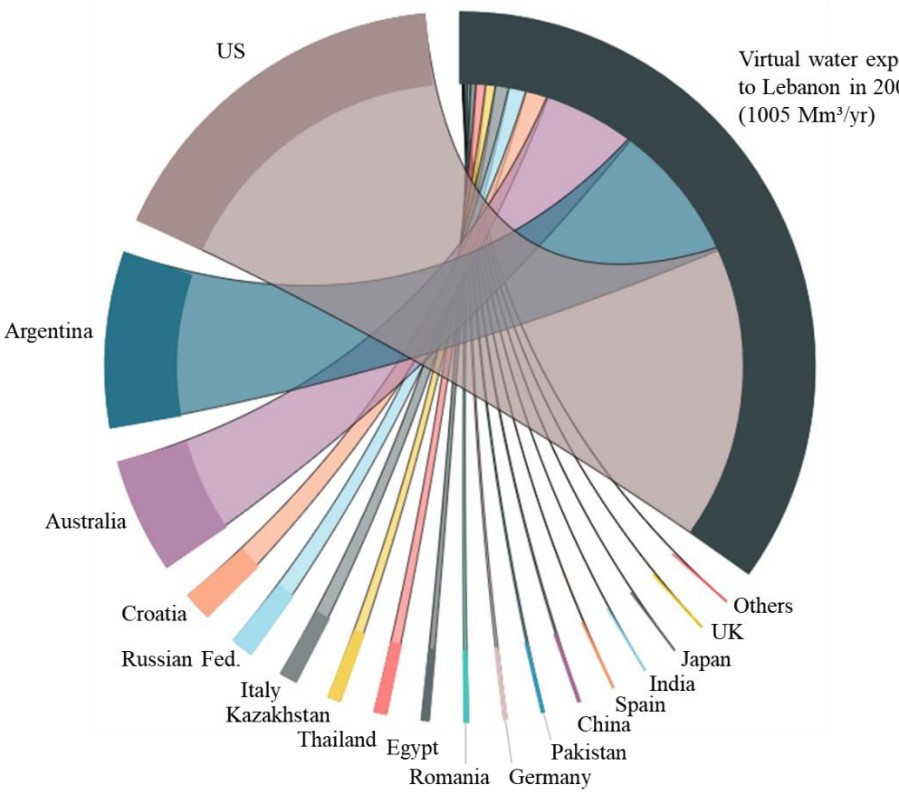

(b) Lebanon in 2000


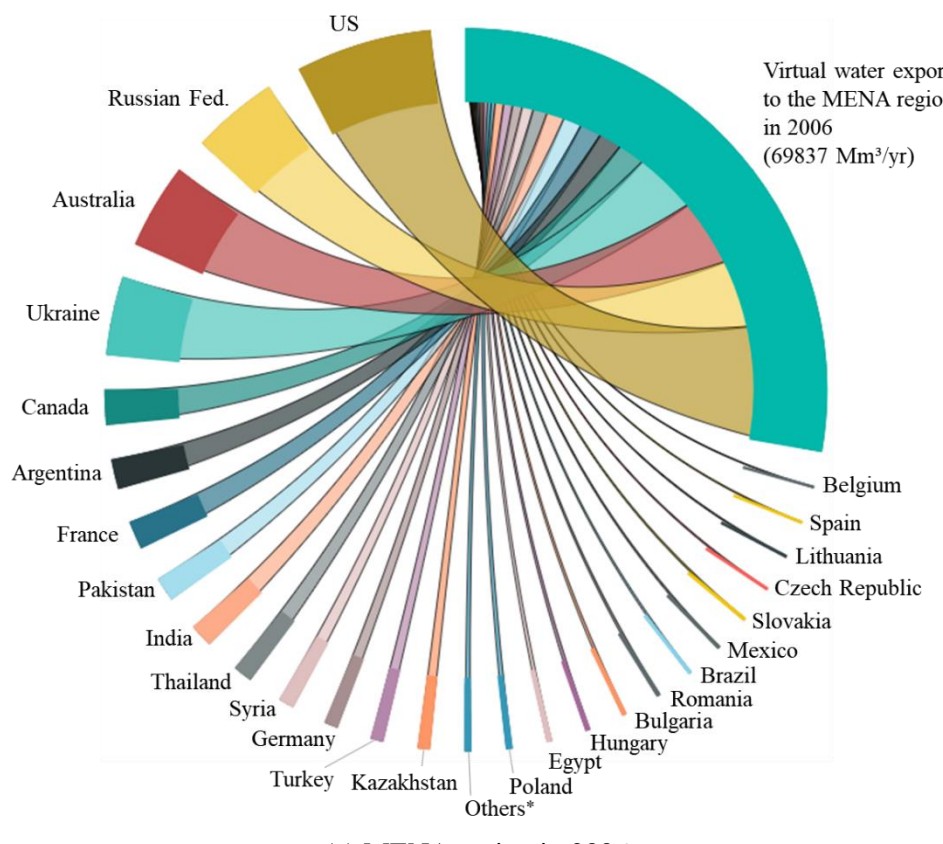

(c) MENA region in 2006

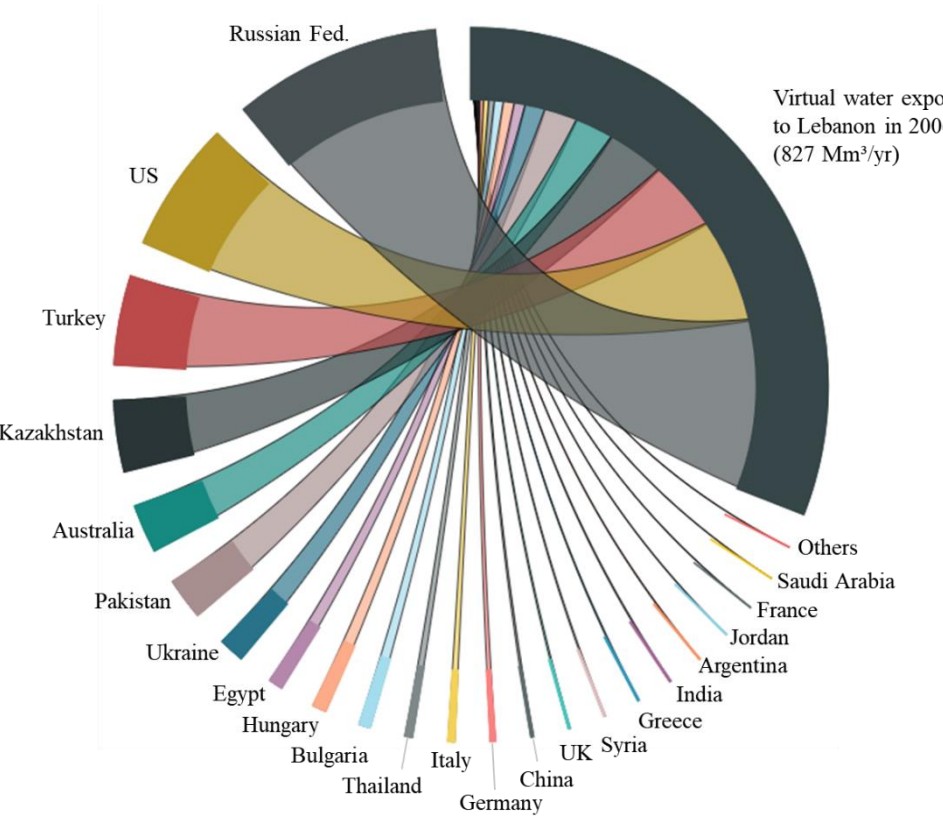

(d) Lebanon in 2006


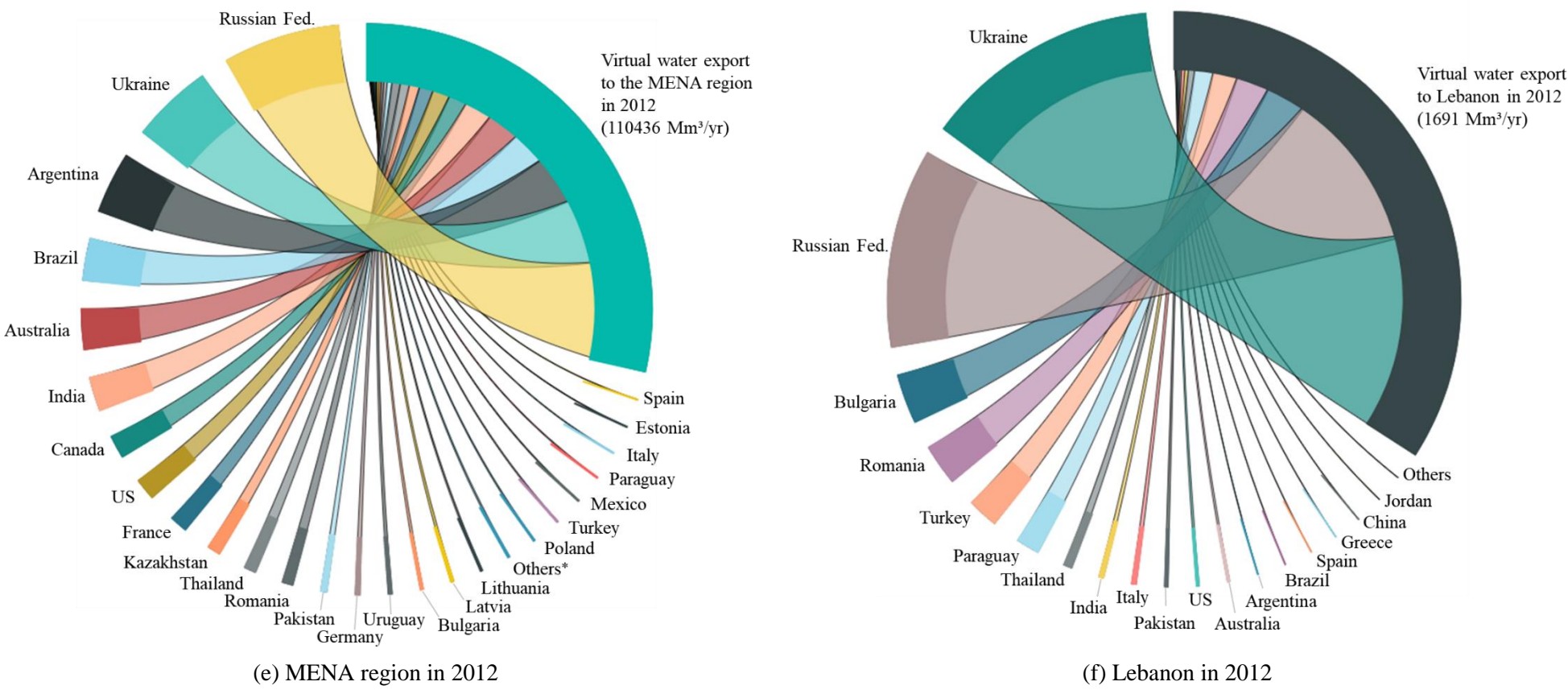

(e) MENA region in 2012          (f) Lebanon in 2012

**Figure 4.** Virtual water imports at the MENA region and Lebanon in 2000, 2006, and 2012. Others indicate the countries who export less than 100 Mm³/yr to the MENA region or Lebanon

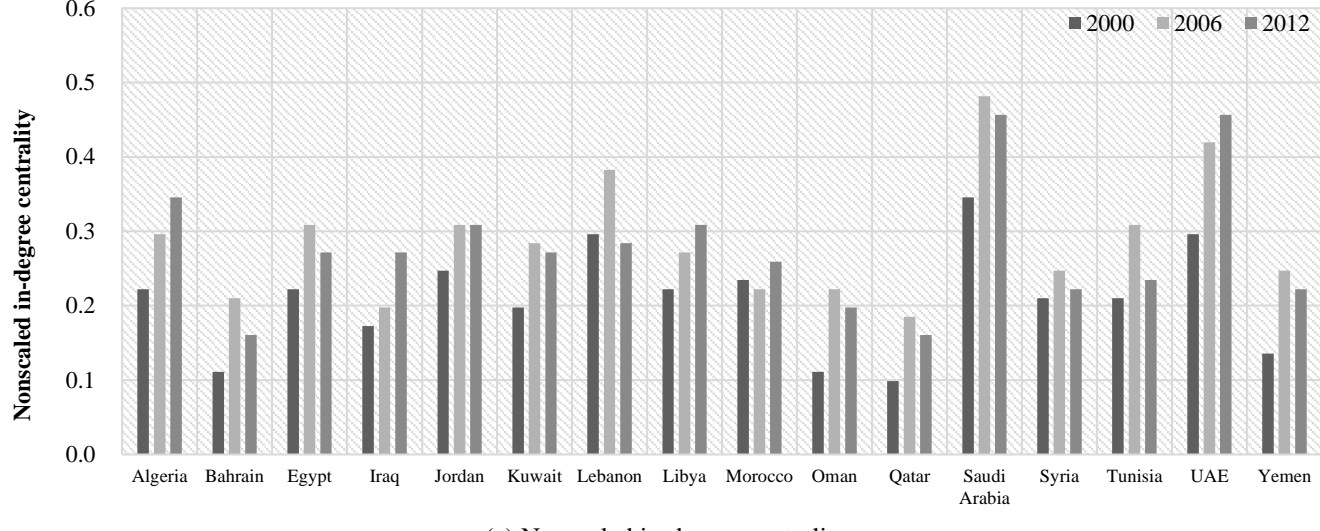

(a) Nonscaled in-degree centrality

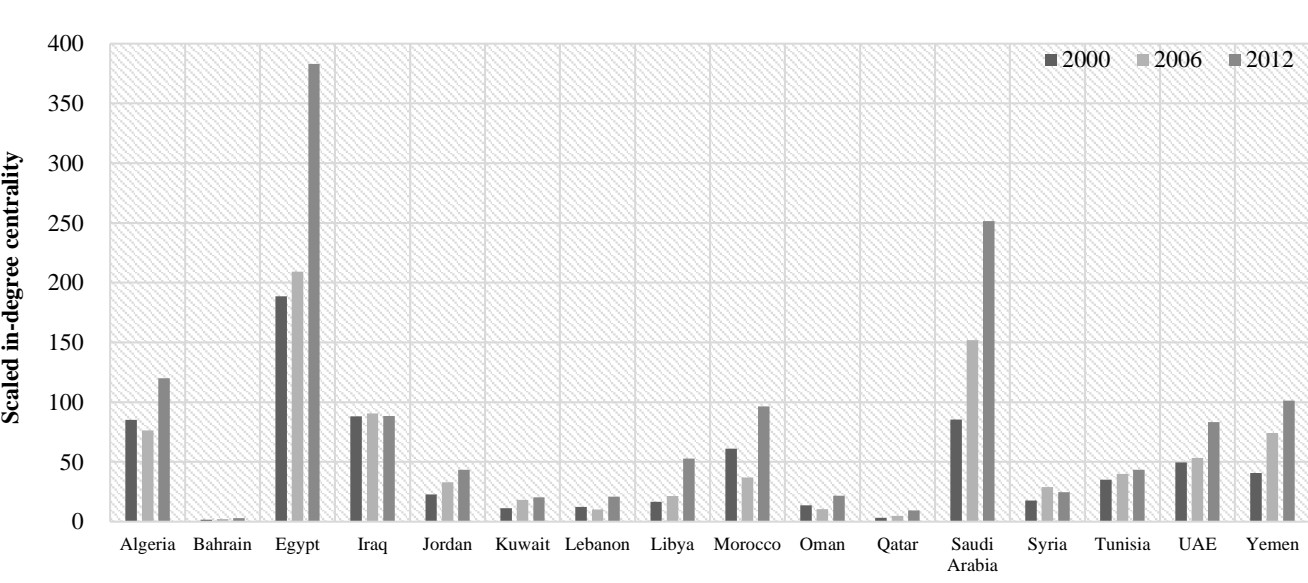

(b) Scaled in-degree centrality

**Figure 5.** Nonscaled and scaled in-degree centralities of each country in the MENA region in 2000, 2006, and 2012

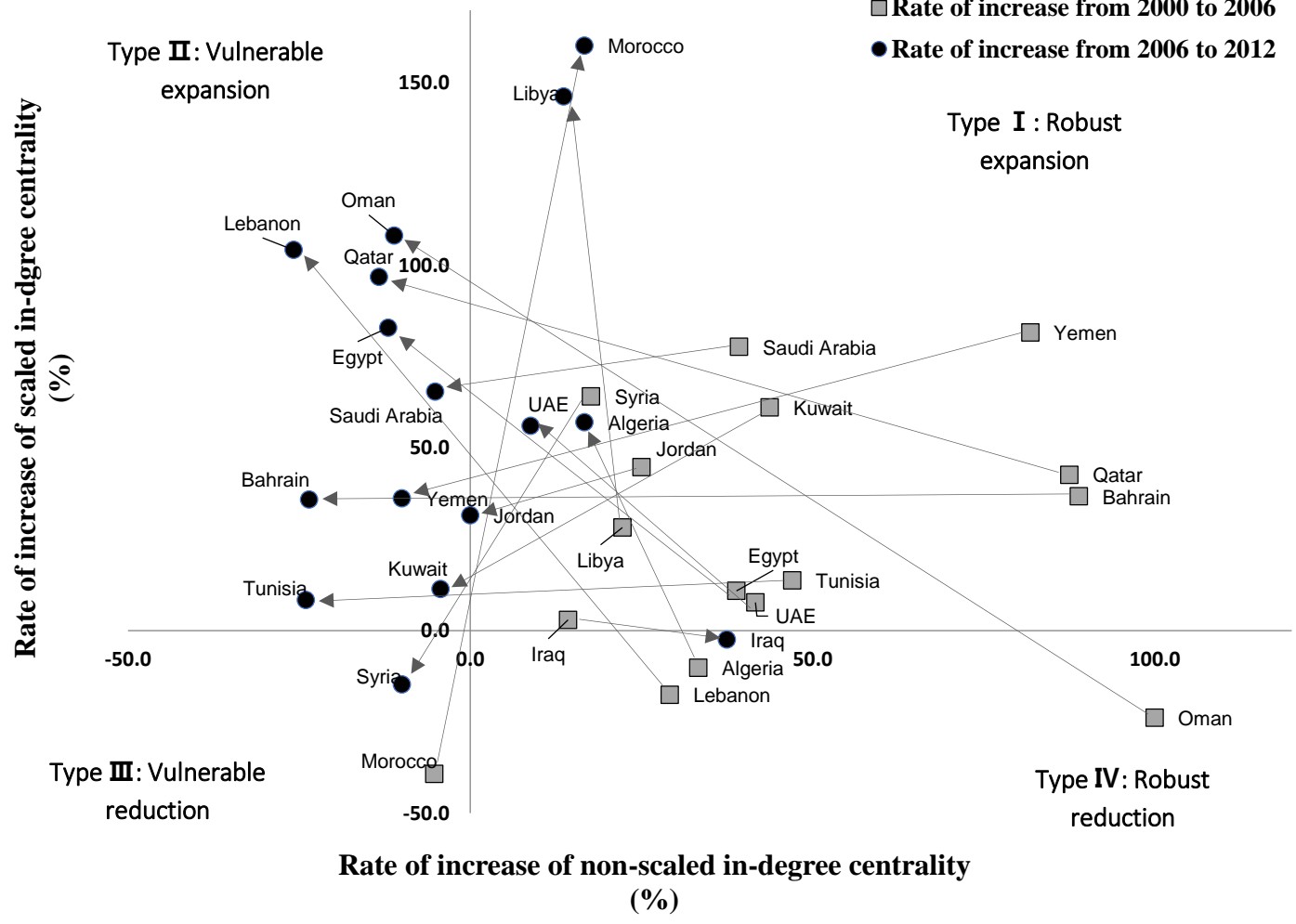

**Figure 6.** Country types in the MENA region according to the changes of nonscaled and scaled in-degree centralities

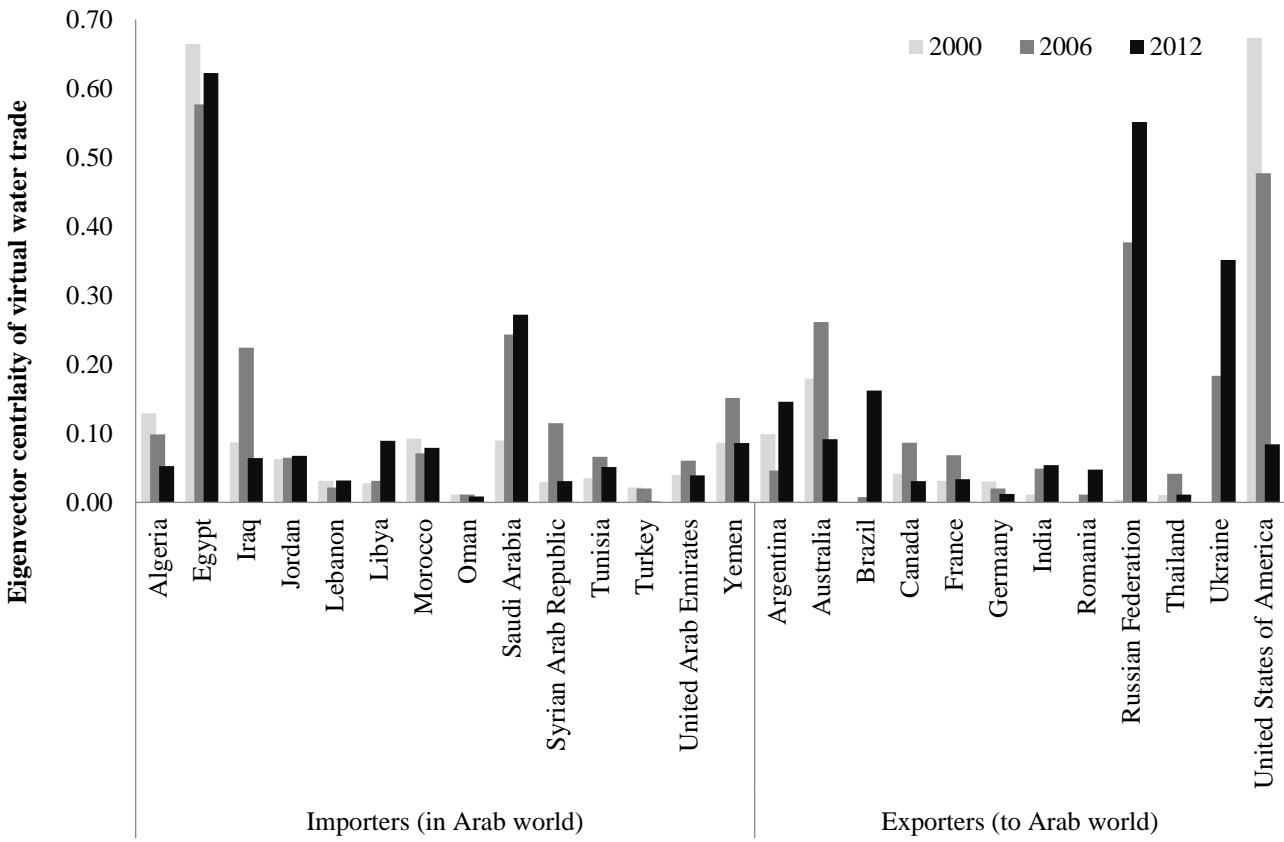

**Figure 7.** Eigenvector centralities of the virtual water trade network in the MENA region in 2000, 2006, and 2012

**Table 1.** Cultivation area, production, and the quantity of crops imported in the MENA region from 2000 to 2012

| Countries in the MENA region | Cultivation area (ha/year)* | | | | | Production (ton/year)* | | | | | Import (ton/year)* | | | | | Internal water resource (10⁹ m³/year)** |
|---|---|---|---|---|---|---|---|---|---|---|---|---|---|---|---|---|
| | Barley | Maize | Wheat | Rice | Sum | Barley | Maize | Wheat | Rice | Sum | Barley | Maize | Wheat | Rice | Sum | |
| ALGERIA | 760,545 | 308 | 1,658,197 | - | 2,419,050 | 1,049,710 | 1,128 | 2,313,464 | - | 3,364,302 | 233,887 | 2,112,527 | 5,363,580 | 47,080 | 7,757,074 | 11.25 |
| EGYPT | 68,103 | 876,153 | 1,180,644 | 625,626 | 2,750,526 | 134,034 | 6,812,845 | 7,549,253 | 6,023,684 | 20,519,816 | 24,805 | 5,073,779 | 8,295,988 | 46,292 | 13,440,864 | 1.80 |
| IRAQ | 914,074 | 128,842 | 1,451,219 | 85,182 | 2,579,317 | 751,099 | 307,682 | 2,009,972 | 232,040 | 3,300,793 | 35,378 | 18,960 | 2,545,919 | 742,394 | 3,342,651 | 35.20 |
| JORDAN | 31,158 | 947 | 20,116 | - | 52,221 | 22,757 | 17,514 | 23,379 | - | 63,650 | 487,593 | 385,936 | 792,508 | 137,442 | 1,803,479 | 0.68 |
| KUWAIT | 1,058 | 290 | 173 | - | 1,521 | 2,191 | 5,855 | 345 | - | 8,391 | 178,432 | 134,373 | 284,684 | 171,451 | 768,940 | - |
| LEBANON | 13,515 | 949 | 45,380 | - | 59,844 | 24,834 | 3,579 | 126,623 | - | 155,036 | 49,278 | 289,707 | 367,370 | 46,087 | 752,442 | 4.80 |
| LIBYA | 191,641 | 1,356 | 165,469 | - | 358,466 | 94,107 | 2,997 | 128,149 | - | 225,253 | 226,317 | 429,407 | 803,545 | 122,579 | 1,581,848 | 0.70 |
| MOROCCO | 2,118,032 | 226903 | 2,910,977 | 5,876 | 5,261,788 | 1,867,670 | 159,127 | 4,200,596 | 36,936 | 6,264,329 | 392,639 | 1,446,836 | 2,994,446 | 13,307 | 4,847,228 | 29.00 |
| OMAN | 1,002 | - | 426 | - | 1,428 | 3,027 | - | 1,432 | - | 4,459 | 35,829 | 99,525 | 288,134 | 118,802 | 542,290 | 1.40 |
| QATAR | 947 | 94 | 15 | - | 1,056 | 2,841 | 1,329 | 34 | - | 4,204 | 33,286 | 3,914 | 47,798 | 87,312 | 172,310 | 0.06 |
| SAUDI ARABIA | 12,279 | 16,689 | 374,414 | - | 403,382 | 68,366 | 86,181 | 1,997,598 | - | 2,152,145 | 6,252,893 | 1,600,081 | 700,703 | 1,009,384 | 9,563,061 | 2.40 |
| SYRIA | 1,313,101 | 53,405 | 1,667,229 | - | 3,033,735 | 817,609 | 211,675 | 4,008,420 | - | 5,037,704 | 393,029 | 1,319,461 | 454,904 | 201,690 | 2,369,084 | 7.13 |
| TUNISIA | 385,189 | - | 722,038 | - | 1,107,227 | 411,431 | - | 1,302,438 | - | 1,713,869 | 407,455 | 737,754 | 1,525,848 | 17,453 | 2,688,510 | 4.20 |
| UAE | 14 | 144 | 18 | - | 176 | 111 | 2,931 | 74 | - | 3,116 | 215,321 | 399,987 | 1,063,996 | 683,336 | 2,362,640 | 0.15 |
| YEMEN | 39,276 | 40,774 | 110,138 | - | 190,188 | 32,248 | 57,329 | 173,437 | - | 263,014 | 2,845 | 343,919 | 2,096,970 | 279,136 | 2,722,870 | 2.10 |

* Average value from 2000 to 2012 provided from FAOSTAT (http://www.fao.org/faostat/)
** Average value from 2000 to 2012 provided from World Bank (https://data.worldbank.org/)


**Table 2.** Water and land footprints of four major crops in the MENA region

| Countries in the MENA region | Water footprint (m³/ton)* | | | | | | | | Land footprint (ha/ton)** | | | |
| | Barley | | Maize | | Wheat | | Rice | | Barley | Maize | Wheat | Rice |
| | Green water footprint | Blue water footprint | Green water footprint | Blue water footprint | Green water footprint | Blue water footprint | Green water footprint | Blue water footprint | | | | |
|---|---|---|---|---|---|---|---|---|---|---|---|---|
| ALGERIA | 2859.0 | - | 964.1 | - | 3290.0 | 65.2 | 1080.8 | - | 0.72 | 0.27 | 0.72 | - |
| EGYPT | 619.2 | 1694.7 | 140.8 | 1078.2 | 214.8 | 903.5 | 59.0 | 1003.1 | 0.51 | 0.13 | 0.16 | 0.10 |
| IRAQ | 3459.7 | 4321.4 | 587.3 | 1812.2 | 3069.2 | 2818.3 | 256.2 | 6574.7 | 1.22 | 0.42 | 0.72 | 0.37 |
| JORDAN | 3167.8 | 320.3 | 126.6 | - | 2267.0 | 988.7 | - | - | 1.37 | 0.05 | 0.86 | - |
| KUWAIT | 929.3 | 2256.3 | 41.2 | 207.9 | 955.4 | 2287.7 | - | - | 0.48 | 0.05 | 0.50 | - |
| LEBANON | 1919.9 | - | 507.6 | 14.4 | 1556.0 | 97.0 | - | - | 0.54 | 0.27 | 0.36 | - |
| LIBYA | 6417.6 | 1808.2 | 1151.1 | - | 4360.2 | 1542.9 | - | - | 2.04 | 0.45 | 1.29 | - |
| MOROCCO | 3692.3 | - | 3541.0 | 3182.9 | 2758.0 | 244.6 | 293.0 | 1278.0 | 1.13 | 1.43 | 0.69 | 0.16 |
| OMAN | 322.9 | 2336.2 | - | - | 842.4 | 1938.5 | - | - | 0.33 | - | 0.30 | - |
| QATAR | 485.6 | 1714.3 | 78.5 | 502.9 | 678.6 | 1626.3 | - | - | 0.33 | 0.07 | 0.44 | - |
| SAUDI ARABIA | 193.6 | 799.8 | 366.6 | 1270.1 | 238.4 | 1093.2 | - | - | 0.18 | 0.19 | 0.19 | - |
| SYRIA | 5084.0 | 41.6 | 347.3 | 1573.4 | 1454.2 | 440.1 | 273.2 | - | 1.61 | 0.25 | 0.42 | - |
| TUNISIA | 3561.1 | 75.1 | - | - | 2375.0 | 71.8 | - | - | 0.94 | - | 0.55 | - |
| UAE | - | - | - | - | 1563.5 | 507.7 | - | - | 0.13 | 0.05 | 0.24 | - |
| YEMEN | 1904.6 | 3234.4 | 1726.2 | 2950.8 | 1804.4 | 2355.5 | - | - | 1.22 | 0.71 | 0.64 | - |

* Water footprint data was referenced by Mekonnen and Hoekstra (2010)
** Land footprint was calculated by crop production and cultivated area provided from World Bank open data (https://data.worldbank.org/)


**Table 3.** The annual water and land savings based on imported crops in the MENA region from 2000 to 2012

| Countries in the MENA region | Water savings (million m³/year) | | | | | | Land savings (thousand ha/year) | | |
|---|---|---|---|---|---|---|---|---|---|
| | Barley | | Maize | | Wheat | | Barley | Maize | Wheat |
| | Green water | Blue water | Green water | Blue water | Green water | Blue water | | | |
| ALGERIA | 669.0 | - | 2,037.2 | - | 17,647.6 | 349.9 | 169.5 | 577.0 | 3,844.7 |
| EGYPT | 15.5 | 42.4 | 714.3 | 5,470.5 | 1,781.9 | 7,495.6 | 12.7 | 652.5 | 1,297.4 |
| IRAQ | 121.1 | 151.3 | 11.2 | 34.4 | 7,814.1 | 7,175.5 | 42.6 | 8.0 | 1,838.2 |
| JORDAN | 1,545.9 | 156.3 | 48.9 | - | 1,797.7 | 784.0 | 668.2 | 20.9 | 682.3 |
| KUWAIT | 165.4 | 401.6 | 5.5 | 27.9 | 272.3 | 652.0 | 86.0 | 6.6 | 142.9 |
| LEBANON | 94.1 | 0.0 | 147.2 | 4.2 | 571.0 | 35.6 | 26.7 | 76.9 | 131.5 |
| LIBYA | 1,450.4 | 408.6 | 493.8 | - | 3,505.6 | 1,240.5 | 460.2 | 194.1 | 1,038.1 |
| MOROCCO | 1,451.1 | - | 5,123.8 | 4,605.6 | 8,257.3 | 732.3 | 445.7 | 2,063.3 | 2,074.8 |
| OMAN | 11.6 | 84.1 | - | - | 242.6 | 558.3 | 11.9 | - | 85.7 |
| QATAR | 16.0 | 56.6 | 0.3 | 2.0 | 32.6 | 78.1 | 11.0 | 0.3 | 21.2 |
| SAUDI ARABIA | 1,210.5 | 5,001.5 | 586.5 | 2,032.1 | 167.1 | 766.3 | 1,123.1 | 309.8 | 131.4 |
| SYRIA | 1,998.0 | 16.3 | 458.1 | 2,075.3 | 661.6 | 200.3 | 631.2 | 332.8 | 189.2 |
| TUNISIA | 1,449.4 | 30.5 | - | - | 3,624.2 | 109.6 | 381.0 | - | 846.0 |
| UAE | - | - | - | - | 1,663.6 | 540.2 | 27.1 | 19.7 | 258.8 |
| YEMEN | 5.7 | 9.7 | 593.8 | 1,015.1 | 3,783.8 | 4,939.4 | 3.7 | 244.7 | 1,331.7 |

* Water and land savings by rice import was not calculated because of the lack of the data of water and land footprints in the MENA region


**Table 4.** The amounts of green and blue water imported in the MENA region from 2000 to 2012

| Countries in the MENA region | Import of green water (million m³/year) | | | | | Import of blue water (million m³/year) | | | | |
|---|---|---|---|---|---|---|---|---|---|---|
| | Barley | Maize | Wheat | Rice | Total | Barley | Maize | Wheat | Rice | Total |
| ALGERIA | 242.0 | 1,883.6 | 5,104.8 | 57.8 | 7,288.2 | 7.8 | 76.6 | 371.1 | 33.5 | 489.0 |
| BAHRAIN | 0.4 | 7.5 | 62.7 | 44.4 | 115.0 | 0.2 | 0.3 | 7.1 | 78.2 | 85.8 |
| EGYPT | 37.3 | 3,798.4 | 15,254.1 | 58.4 | 19,148.2 | 1.1 | 295.6 | 418.6 | 32.5 | 747.8 |
| IRAQ | 33.2 | 16.7 | 4,645.8 | 1,027.8 | 5,723.5 | 2.2 | 1.3 | 153.9 | 404.8 | 562.2 |
| JORDAN | 656.8 | 364.2 | 1,483.9 | 81.2 | 2,586.1 | 20.8 | 20.8 | 84.5 | 115.0 | 241.1 |
| KUWAIT | 257.0 | 159.1 | 557.7 | 211.6 | 1,185.4 | 9.7 | 2.3 | 10.2 | 138.1 | 160.3 |
| LEBANON | 84.7 | 211.0 | 749.5 | 30.0 | 1,075.2 | 2.3 | 25.6 | 18.9 | 36.0 | 82.8 |
| LIBYA | 359.6 | 408.9 | 1,245.4 | 56.0 | 2,069.9 | 8.4 | 26.8 | 75.3 | 99.7 | 210.2 |
| MOROCCO | 318.6 | 1,383.2 | 3,345.0 | 8.9 | 5,055.7 | 12.1 | 46.1 | 118.8 | 20.4 | 197.4 |
| OMAN | 52.7 | 123.2 | 470.8 | 107.6 | 754.3 | 5.4 | 4.1 | 67.8 | 201.3 | 278.6 |
| QATAR | 50.9 | 6.4 | 76.4 | 77.6 | 211.3 | 2.4 | 0.3 | 19.1 | 146.9 | 168.7 |
| SAUDI ARABIA | 8,154.5 | 1,521.4 | 974.0 | 1,225.9 | 11,875.8 | 324.3 | 68.9 | 70.8 | 696.0 | 1,160.0 |
| SYRIA | 556.4 | 947.3 | 900.0 | 120.8 | 2,524.5 | 12.8 | 90.2 | 17.8 | 165.6 | 286.4 |
| TUNISIA | 409.8 | 611.7 | 2,507.7 | 27.8 | 3,557.0 | 16.0 | 40.7 | 73.9 | 11.6 | 142.2 |
| UAE | 315.7 | 465.8 | 1,671.8 | 859.5 | 3,312.8 | 28.5 | 14.3 | 249.3 | 612.5 | 904.6 |
| YEMEN | 3.1 | 406.1 | 3,597.3 | 392.7 | 4,399.2 | 1.6 | 8.2 | 247.3 | 220.8 | 477.9 |
