# Peer review of "Assessment of food trade impacts on water, food, and land security in the MENA region"

_Hydrology and Earth System Sciences, 2018_

## Referee Comment (RC1) · M. Walther (Referee) · 12 Oct 2018

M. Walther (Referee)

marc.walther@ufz.de

Received and published: 12 October 2018

Comments on the manuscript Assessment of food trade impacts on water, food, and land security in the MENA region submitted to Hydrology and Earth System Sciences by Lee, Mohtar and Yoo

The manuscript analyzes the virtual water trade network for import and export of blue and green water of four crop types for the MENA region. The analysis is carried out with a number of indices (water footprints, in-degree centralities) and eigenvectors and utilizes data from previous studies. The main conclusions highlight food sector securities for the countries of the MENA region and their temporal development over the years 2000 - 2012. I consider this study to be a timely and viable approach. I acknowledge

the discussion of water availability through food security and land availability.

Generally, the methods are concisely described, figures are mostly meaningful, tables support the text, yet both of the two latter can be enhanced. There are some occassions where statements are unnecessary or unproven which should be revised (see specific comments below). The introduction cites many valid references, but I think that the manuscript should discuss many more. I had a very quick search for "food nexus MENA" in ScienceDirect which brought the following results that definitely should be discussed:

F. Saladini, G. Betti, E. Ferragina, F. Bouraoui, S. Cupertino, G. Canitano, M. Gigliotti, A. Autino, F.M. Pulselli, A. Riccaboni, G. Bidoglio, S. Bastianoni, Linking the water-energy-food nexus and sustainable development indicators for the Mediterranean region, Ecological Indicators, Volume 91, 2018, Pages 689-697, ISSN 1470-160X, https://doi.org/10.1016/j.ecolind.2018.04.035.

Mohammad Al-Saidi, Diana Birnbaum, Renata Buriti, Elena Diek, Clara Hasselbring, Andres Jimenez, Désirée Woinowski, Water Resources Vulnerability Assessment of MENA Countries Considering Energy and Virtual Water Interactions, Procedia Engineering, Volume 145, 2016, Pages 900-907, ISSN 1877-7058, https://doi.org/10.1016/j.proeng.2016.04.117.

Afreen Siddiqi, Laura Diaz Anadon, The water—energy nexus in Middle East and North Africa, Energy Policy, Volume 39, Issue 8, 2011, Pages 4529-4540, ISSN 0301-4215, https://doi.org/10.1016/j.enpol.2011.04.023.

Lanouar Charfeddine, Zouhair Mrabet, The impact of economic development and social-political factors on ecological footprint: A panel data analysis for 15 MENA countries, Renewable and Sustainable Energy Reviews, Volume 76, 2017, Pages 138-154, ISSN 1364-0321, https://doi.org/10.1016/j.rser.2017.03.031.

Khalil Lezzaik, Adam Milewski, Jeffrey Mullen, The groundwater risk index: Devel-

**HESSD**
opment and application in the Middle East and North Africa region, Science of The Total Environment, Volumes 628–629, 2018, Pages 1149-1164, ISSN 0048-9697, https://doi.org/10.1016/j.scitotenv.2018.02.066.

I am sure, there are many more, but I tend to leave this research to the authors. I also miss a discussion of the analysis that is solely based on the data from the last years with different societal, political and environmental aspects; currently, the manuscript only shows the changes in food supply security and interprets the results without considering the bounding conditions for the MENA countries, which strongly differ. Finally, I think that especially the conclusions section should be more detailed and overhauled - currently, this is only a collection of vague statements, but the analysis and the presented results show much more potential of detailed conclusions; for example, the results could be synthesized for all the countries of focus in a comparable way.

If the authors can address the issues above (broader coverage/discussion of relevant publications, country-specific aspects influencing food trade, clearer conclusions) together with the specific comments listed below, I suggest the editors to accept the manuscript for publication. If the authors consider my comments to be valuable, I would be available for a second revision.

**Specific comments**

Line 27: Please add adequate sources to state that the primary resource gaps will grow. (Maybe, the ones in L69 will work?)

L29: What do you mean by saying "the food portfolio [...] has been complicated by and increased degree of risks..."?

L30: Please provide sources that the MENA region shows tendencies for an inability to satisfy needs with domestic production.

L32: You say that (food) trade has been understudied - one might argue that as trade is a central part of food security (which you likewise support), it is quite well understood
by the relevant trading actors.

L29, 33: I think, MENA & VWT (and all other abbreviations) should be defined in the text (not in the abstract).

Concerning the meaning of VWT: if a product uses 1000 l/kg water to be produced in one region, it might have a much more severe impact in an arid climate than in a humid one (you cannot grow coffee in Lybia, but in Chile). If the value is to be interpreted locally, doesn't it lose its meaning and transferability?

L56: You say that Fader et al (2011) show water savings of 263 km3/a due to beneficial agricultural production in other countries; does this calculation include the additional costs that arise from transport? Additionally, I am wondering how much the import of exotic products to western countries (an unnecessary trade in comparison to the import of basic crop products to arid countries) contributes to in the large savings (17 billion m3 blue water, L65) of global extent?

L111: please add units to WS/LS.

L114/115: Two sentences starting with "In addition" - please revise. I also do not understand the meaning of "In addition, each variable is dependent on local characteristics."

L118: If you irrigate a crop with rain harvested water, either directly as water is used from the reservoir or indirectly as the reservoir water is used for enhanced groundwater recharge, is this blue or green water?

L120: "Thus, the study for national water footprint should be executed for each country, basin, or specific area; however, this was outside the scope of the current study." - this sentence is unclear to me, especially the first part: what is the difference between "national" and "country"? For which regional unit did you carry out your study?

Can you please name the countries of the MENA region that you studied in the beginning, e.g. around L87ff?
L127: What is the "limited water footprint"?

Table 1: - Do I understand it correctly that the information in Table 1 is taken from Mekonnen and Hoekstra (2010)? If so, please add this information in the table caption. - Please add the time period of the data in the caption. - Can you please explain why the blue water footprint is larger than the green water footprint? Why does a plant need less rainwater than groundwater? - Which footprint did you use to calculate the land footprint?

Table 2: - Again, please add the source of this data in the caption. - This data is shown for the years 2000-2012; I assume there are all mean values - please add this information. - If these are mean values, what was the standard deviation of the data? Is there a trend in the data? - Can you please add how this data was acquired and certain this data is? - Can you add a row showing the sums of the individual columns?

L154: It is good that you list previous network-based approaches that investigated VWT structures, but you should not only mention the citations and rather shortly summarize their works and how your work contributes to this.

Equations 6 & 7: - is "j" in the sums as the starting counter equal to 1? I think, the usage of "j" is misleading, as it also refers to exporting countries. - is N (total number of countries) constant for all i (importing countries)? What if a country i only trades with one other country, i.e. N = 1; then, the equation gives a division by zero, correct?

Equation 7: Why is the SInDC not related to the total volume of virtual water traded but to the number of total number of countries?

L172 & 173: I think, it should be "high levels" and "low levels".

Eq 8: What is \_alpha\_ij?

L196ff - Please revise this paragraph: - The first sentence rather belongs to a summary, after you showed results, but you did not at this place in the manuscript. - The second sentence is given without reference/citation. - The third sentence contradicts the first

**HESSD**
two sentences. - The fourth sentence does not state whether Egypt imports from MENA countries or somewhere else. - The fifth sentence is not justified by the one example you state. - I also do not understand the intension of this paragraph, what do you want to convey here? Even the following sentence in L202 starts with "however" as if you wanted to say "but I actually want to talk about something else".

L206: "This means that the contribution of import of barley, maize, and wheat on water security in Saudi Arabia was significant." - how do you come to this conclusion?

A general comment: for example in L209, you state that Egypt would suffer from water shortage if the exporting countries banned wheat export to Egypt. I think that this is only partly true, i.e. only in those cases where the respective crops would actually grow in the individual countries. Considering rice, for example: I am sure that none of the MENA countries would be able to grow this crop even if the virtual water equivalent would be available. Please elaborate on this comment.

L208: The statement of 1.8 billion m3/a water available for Egypt is missing a source.

L210: "The crop import could result in a large amount of land savings." - this is an unnecessary statement. Likewise in L215: "These results can elicit useful information for analyzing the trade-off between food and water-land securities in the MENA region in terms of sustainable development."

L210ff: "In Saudi Arabia, land savings based on the import of barley, maize, and wheat, amounted to 1.6 million ha/year, and Lebanon was also strongly influenced by the impact of crop import on land savings. For example, approximately 0.24 million ha could be saved by crop imports, comprising 36% of the agricultural area in Lebanon, that indicates that the crop trade in Lebanon has significant benefits in terms of land resources compared to water resources." - please revise and do not mix two different countries in different sentences.

L216: What do you mean by this: "However, water saving indicates the virtual water
saving, and sometimes it is larger than the total water resources in some countries. "

L216/217: "However" twice as starting word.

L217: "However, these results showed that the increase of food security is accompanied by numerous water requirements in the MENA region." - I do not understand this. Please revise.

L218ff: "Additionally, the saved land is not always suitable for agricultural areas." - The "saved land", i.e. the equivalent required area to grow imported crops, is probably not available. Do you have information on this?

"Some crops are required for the specific type of land, ..." - It is rather the other way: you require a specific soil for this or that crop.

"...and the productivity is also different based on soil." - Do you mean "the productivity is varies with different soils"?

"Even if we can save land..." - Why do you think, the reason to import is to save land? - Why do you write "we"?

"...there is the limitation for considering the land saving as an agricultural land saving in accordance to this study." - What do you mean by this?

Table 3: - Please check for unnecessary line breaks (eg. Saudi Arabia, Blue water, Barley). - Do I understand it correctly that table 3 shows the results from the product of water footpring (table 1) and the annual import (table 2)? If so, how could you fill the gaps for the water footprint in blue water barley and green water maize? -> Oh, I see you wrote "0" for partly - please correct this and write "-".

Section 3.1 should be shortened; often, statements are given that are unnecessary, unproven or uncited. The information from table 3 can and should be offered in a much more compact way.

L227: Are the numbers for annual water import average values?

**HESSD**
Fig 1: - The grey scale (ie the total water import) uses uneven separating numbers and unequal intervals; I suggest to use even numbers (e.g. 1500 - 15000 instead of 1495 - 15410 for the first green water import interval) and evenly spaced intervals. - I cannot read the number in the legend for annual water import - Some pie charts are very small (Qatar, Oman, Bahrain, Lebanon) - Why do the pie charts vary in size?

Table 3 vs 4: I do not understand the difference between "water savings due to imported crops" (table 3) and "imported water" (table 4) - can you please explain this difference and describe why both values are different?

Section 3.2.2 / figure 3: how could you determine which water (blue or green) was used to grow the crops in the exporting countries?

- Fig. 3: Why do you give the numbers here in Gm3 while all other volumes are given as volume / time (Mm3/y)? I suggest to be consistent for comparability especially with such large numbers which are hard to imagine.
- Fig. 4: the width of the countries should be identical in a and b; please correct this (I assume, the y-axis numbers need to have the same number of digits, then the figures should show the same size).
- Fig. 5: This is a very nice interpretation, but I have a suggestion: you could combine a and b and connect the individual countries' marks with arrows; currently, one has to search for a long time before a country's performance can be compared.
- Fig. 6: Please check for non-discribed countries and/or add them to "others". The numbers of the individual eigenvectors are too small and cannot be read. Can you show this figure also for the whole MENA region? Or in other words: why did you choose Lebanon here? Is the figure similar for the other countries?

L359: If you write "Since the introduction of the virtual water concept, various studies have been conducted to quantify the volume of the VWT." you should provide proper citations and describe how you contribute to an extension of their findings.
L361: As above, the statement "The amount of imported virtual water is regarded as the most important factor in determining water and food security," should be backed up by citations or proof.

L364: "...the interlinkages of key natural resource sectors and the improved production efficiency are considered a win—win strategy for environmental sustainability..." - I do not understand why you address production efficiency here; that was not part of you previous analysis. Can you please explain this?

L368: "Thus, decisions made in one sector typically impact the other sectors." - I think that this statement here does not belong to your core message of the paper: you never discuss / analyze how different sectors influence each other. You also do not show how virtual water or changes in virtual water fluxes may influence whatever sector.

L372: "...policy makers can benefit..." - how should they benefit? What would be the key parameter policy makers can use? How should they decide on the future if your study is only based on the analysis of data from the past? Also: you compared the different countries of the MENA region among each other and derived values for SInDC and NSInDC. The comparison is thus only a qualitative comparison. How should a single country decide now whether its food import strategy generally is stable? Finally: considering political differences in the MENA region, do you think that any singular country or a coalition of countries could use your evaluation to increase its food stability?

---

## Referee Comment (RC2) · Biggs (Referee) · 21 Dec 2018

This paper analyzes the virtual water trade (VWT) in the MENA (Middle East and North Africa) region with data from 2000 to 2012. They use previously-compiled data and analyze the importance of VWT in MENA countries' food supply. More novel, they determine the degree of dependence of MENA countries on single exporters, and the centrality of different exporters for the VWT throughout MENA. The study usefully visualizes and analyzes existing VWT data, and would be a good addition to the literature on VWT and how to analyze it. I have mostly minor corrections for language (see attached pdf) but also some questions about the results from water footprints, and suggestions for further interpretation of some data, including whether there is historical precedent for countries that depend on a few exporters to be more vulnerable to food

shortages from embargoes or other trade policy.

The water footprints of a given crop vary widely by country: for barley, green WF ranges from 193.6 to 6417.6 m3/ton. Adding together green and blue still gives a very wide range: ~8200 m3/ton in Libya vs 1000 m3/ton in Saudi Arabia. Are these numbers and their spatial variability realistic? Is it possible that producing barley in Libya consumes 8 times as much water as in Saudi Arabia? I don't imagine that potential ET varies that much over the region. Is the very wide range in WF because yields are so much higher in Saudi Arabia, but water consumption is assumed to be independent of yield? Some explanation is needed.

I found the methods description for Eigenvector centralities confusing (L176-193). Please rewrite for clarity.

Some numbers are claimed to be significant, but without context. For example, Saudi Arabia saves 2 billion m3 per year by importing barley. Is that big number? Compared to what?

The authors correctly note that it is important to identify countries that rely on only a few exporters. I'm not so sure that this means that countries with high dependence on one exporter should re-evaluate their policy, since I don't know enough about international trade strategy. Is there literature that can show that, historically, countries that rely on a single exporter are vulnerable to food sanctions? Can the authors cite historical precedent? Also, I found the shift in exporting countries from the US and Australia to other nations of potential importance to explain, both its causes and consequences. Is there more you can say about that in the paper?

Please also note the supplement to this comment:
https://www.hydrol-earth-syst-sci-discuss.net/hess-2018-398/hess-2018-398-RC2-supplement.pdf

398, 2018.

[revised manuscript text omitted]

---

## Author Comment (AC1) · 13 Jan 2019

Dear reviewer and editor,

thank you for considering the manuscript for publication in the HESS and in-depth review of the manuscript. We believe food trade bring important impacts on water-food-lands management in the MENA region. Therefore, this study focused on quantifying domestic water-lands savings by food trade, and we analyzed the virtual water trade in terms of volume and connectivity. In reviewer's comments, we identified the main critiques directed towards the weak explanation of the situation of the MENA region, limitations and contribution of this study, and proposed methodology. We have made substantial changes to the manuscript to improve upon these points. For example, in

revised manuscript, we added more reference studies for identifying the situation of the MENA region, and clarify the limitation of this study in terms of policy application for example, only historical data use and lack of geopolitical issues. In addition, we rewrote the methodology of eigenvector centrality with more references, and added more explanation about the difference between water saving and virtual water import. We attached revision note and revised manuscript (zip file) in supplement, and you will find an overview of changes and a point-by-point reply to specific comments. We appreciate again your thoughtful comments, and look forward to hearing your reply.

Kind regards, on behalf of all co-authors, Sanghyun Lee

Please also note the supplement to this comment:
https://www.hydrol-earth-syst-sci-discuss.net/hess-2018-398/hess-2018-398-AC1-supplement.zip

---

## Author Response (AR1)

**Reply to the referees for hess-2018-398**

Article title: Assessment of food trade impacts on water, food, and land security in the MENA region
Authors: Sang-Hyun Lee, Rabi H. Mohtar, Seung-Hwan Yoo

Dear reviewers and editor, thank you for considering the manuscript for publication in the HESS and in-depth review of the manuscript. We believe food trade bring important impacts on water-food-lands management in the MENA region. Therefore, this study focused on quantifying domestic water-lands savings by food trade, and we analyzed the virtual water trade in terms of volume and connectivity.
In reviewer's comments, we identified the main critiques directed towards the weak explanation of the situation of the MENA region, limitations and contribution of this study, and proposed methodology. We have made substantial changes to the manuscript to improve upon these points. For example, in revised manuscript, we added more reference studies for identifying the situation of the MENA region, and clarify the limitation of this study in terms of policy application for example, only historical data use and lack of geopolitical issues. In addition, we rewrote the methodology of eigenvector centrality with more references, and added more explanation about the difference between water saving and virtual water import. On the next pages you will find an overview of changes and a point-by-point reply to specific comments.
We appreciate again your thoughtful comments, and look forward to hearing your reply.

Kind regards, on behalf of all co-authors,
Sanghyun Lee

**Overview of changes**

We tried to revised the paper with your comments. Please find the overview of changes and point-by-point reply to specific comments. In terms of general comments, first we revised the introduction by adding more references about the situation of the MENA region, and added more explanation about the differences between water saving and virtual water import. In addition, we added more limitations in terms of spatial and temporal issues of VWT, and mentioned contribution and future works in conclusions. Finally, we checked entire manuscript and revised some paragraph and typo.

1. **We revised the introduction by adding more references about the situation of the MENA region.**
   **Page 1: Line 27– Line 36**
   Food security and water scarcity are urgent socio-economic and environmental issues in the Middle East and North Africa (MENA) region (Saladini et al., 2018), which are highly interlinked, and Water-Energy-Food Nexus has been suggested as a proper and integrated approach for resource management (Bazilian et al., 2011; Rasul, 2014; Mohtar and Daher, 2014; Lee et al., 2018). For example, food security in the MENA region has become complicated by increased risks owing to the geopolitical challenges and inability to satisfy needs with domestic production because of the lack of adequate arable land and water resources (Rastoin and Cheriet, 2010). In addition, food imbalance in the MENA region is forecast to reach 60 % in 2050 and food security in MENA region could be extremely compromised (Rastoin and Cheriet, 2010). Climate change could lead to more frequent occurrence of extreme climatic events in Mediterranean region, accompanying 50 % decrease of agricultural production by the end of the century (Porter et al., 2014). In particular, water saving through food trade can be suggested as a solution for mitigating groundwater depletion in the MENA region (Lezzaik et al., 2018).

2. **We added more explanation about the differences between water saving and virtual water import.**
   **Page 4: Line 133– Line 144**
   Food import is also related to domestic water and lands savings. In particular water saving has a different meaning from virtual water import. For example, Saudi Arabia imported wheat from various exporters and virtual water import indicates the sum of the products obtained from multiplying the quantity of imported wheat by the respective water footprint of each exporter. However, water saving indicates the amount of water needed to produce the same quantity of imported products domestically. Therefore, water saving by wheat import in Saudi Arabia is estimated by multiplying the quantity of imported wheat with the water footprint of wheat in Saudi Arabia.
   In this study, we applied green and blue water footprints of crops in each country in the MENA region, as shown in Table 1. However, the availability of water footprint data in the MENA region was limited in some cases. For example, the water footprint of wheat was available in all countries except for Bahrain. Lands saving has the same implication as water savings, thus we calculated lands saving using land footprint of each country in the MENA region, as shown in Table 2. The land footprint indicates the land requirement for producing 1 ton of crops, and it was calculated based on the harvest area and crop production data collected from FAOSTAT.

3.  **We revised the entire part of section 3.1 to clarity the results.**
    **Page 6: Line 219– Line 238**
    This study considered trade-offs between food security and food trade in terms of national resource management. For example, the increase of domestic food products instead of imports of them could be one policy for food security but additional water and land for domestic products would be considered at the same time. In other words, food imports could contribute domestic water and land management, therefore, we estimated the national water and land savings by importing crops as shown in Table 3. In Saudi Arabia, blue water savings by barley, maize, and wheat imports were estimated to 5.0, 2.0 and 0.8billion m³/year, respectively. In comparison to the internal water resource of Saudi Arabia which is 2.4 billion m³/year as shown Table 1(World Bank, 2014), the water saving through import of barley, maize, and wheat could be considered as significant amount in Saudi Arabia. In the case of Egypt, most of the water saving occurred based on the imports of wheat and maize. Approximately 7.5 billion m³/year of blue water was saved by importing wheat. Specifically, the internal water resources in Egypt are only 1.8 billion m³/year (Table 1), therefore, water scarcity could be an issue for food security policy in Egypt. Lebanon was strongly influenced by the impact of crop import on land savings. Approximately 0.24 million ha could be saved by crop imports, comprising 36% of the agricultural area in Lebanon, that indicates that the crop trade in Lebanon has significant benefits in terms of land resources compared to water resources.

    Food imports could be regarded as a negative factor in food security, and it is obvious that food security would accompany water and lands for domestic food products. These results showed that food imports could bring positive impacts on numerous water and lands savings in the MENA region. However, there are limitations of these results. First, water saving estimated in this study was based on the hypothetical situation that meat there were no international trade situation, and sometimes it was larger than the internal water resources in some countries such as Saudi Arabia and Egypt. Additionally, some crops are required for the specific type of climate but this study assumed that MENA region was suitable for cultivating maize, wheat, barley, and rice.

4.  **In previous version, virtual water import diagram of only Lebanon was showed as a case but in revised version, we added virtual water import diagram of total MENA region and added explanation in section 3.3.**
    **Page 8: Line 289 –Line 297**
    From 2000 to 2012, both the volume and connectivity of VWT was changed. For example, the virtual water imported in the MENA region slightly increased and the VWT was distributed with more exporters in 2006, as shown in Figure 4. However, the volume of virtual water imported in the MENA region was increased more than 50 % from 2006 to 2012 but the distribution of VWT seemed to consistent. In case of Lebanon, VWT in Lebanon was strongly dependent on the USA, Argentina, and Australia. However, Lebanon expended the VWT in 2006 and Russian Federation, Turkey, and Kazakhstan, contributed to virtual water imports in Lebanon, as shown in Figure 4. Accordingly, the structure of VWT in Lebanon approached a distributed network. However, the VWT in 2012 showed that it was dominated by Ukraine and Russian Federation, though Lebanon imported more virtual water in 2012 than 2006.
    **Figure 4.** Virtual water imports at the MENA region and Lebanon in 2000, 2006, and 2012

5. **We added more limitations in terms of spatial and temporal issues of VWT.**
   **Page 9: Line 358– Line 363**

   Third, there are spatial and temporal issues of VWT in the study. The VWT could be affected by geopolitical issues such as topography, and distances between importers and exporters. For example, the changes of exporting countries in the MENA region could be related to energy use for transporting products, thus trade policy should consider the economic benefit or cost of transportation. Therefore, the VWT should be discussed with geopolitical issues such as benefit and cost of transportation. In addition, VWT and water-lands savings by food trade in this study were calculated based on historical database, thus it was difficult to apply the results to future policy.

6. **We mentioned some future works in conclusions, for example, relationship between trade and energy part (energy use for transportation and food production).**
   **Page 10: Line 383 – Line 398**

   In summary, this study showed that the significant water in comparison to internal water resource could be saved by food trade in the MENA region, and policy makers can benefit by considering both the quantitative impacts of VWT and the structural changes of VWT, such as vulnerable expansion (or reduction) in the MENA region. For example, when a country in the MENA region set a plan for increasing food security, this country first should identify the amount of water and land savings that can be achieved by food import, and consider the trade-off between food security and food import. In addition, the stable trade could be a component for stable food supply in the MENA region, thus this study contributes to the understanding of the dependency on each trade partner for countries in the MENA region and can help with setting the food trade policy in terms of extension (or reduction) of trade partners and increase (or decrease) in volume of trade.

   However, this study only focused on food trade and water-land savings, thus energy part was not considered. The MENA region represents an extreme case globally in terms of water and energy resources, for example, 66% of the world's known crude oil reserves, but only 1.4% of the world's fresh water supplies is attributed to the region (Khater, 2001). The increase or decrease of water withdrawal for irrigation is related to the energy used for water extraction such as pumping surface or ground water. For example, 5 % or more of the total electricity consumption can be attributed to water pumping in Saudi Arabia (Siddiqi and Anadon, 2011). Energy use for food production and water supply could be the main factor in integrated resource management in the MENA region, and the lack of energy part was a limitation in this study.

7. **We checked entire manuscript and revised some paragraph and typo. Please find them in the revised manuscript.**

**Point-by-point reply to specific comments**
**Referees #1**

**A general comment**

**Generally, the methods are concisely described, figures are mostly meaningful, tables support the text, yet both of the two latter can be enhanced. There are some occassions where statements are unnecessary or unproven which should be revised (see specific comments below). The introduction cites many valid references, but I think that the manuscript should discuss many more. I had a very quick search for "food nexus MENA" in ScienceDirect which brought the following results that definitely should be discussed:**

**I am sure, there are many more, but I tend to leave this research to the authors. I also miss a discussion of the analysis that is solely based on the data from the last years with different societal, political and environmental aspects; currently, the manuscript only shows the changes in food supply security and interprets the results without considering the bounding conditions for the MENA countries, which strongly differ.**

**Finally, I think that especially the conclusions section should be more detailed and overhauled - currently, this is only a collection of vague statements, but the analysis and the presented results show much more potential of detailed conclusions; for example, the results could be synthesized for all the countries of focus in a comparable way. If the authors can address the issues above (broader coverage/discussion of relevant publications, country-specific aspects influencing food trade, clearer conclusions) together with the specific comments listed below, I suggest the editors to accept the manuscript for publication. If the authors consider my comments to be valuable, I would be available for a second revision.**

➔ We tried to revised the paper with your comments. Please find the overview of changes and point-by-point reply to specific comments. In terms of general comments, first we revised the introduction by adding more references about the situation of the MENA region, and added more explanation about the differences between water saving and virtual water import. In addition, we added more limitations in terms of spatial and temporal issues of VWT, and mentioned contribution and future works in conclusions. Finally, we checked entire manuscript and revised some paragraph and typo. Please find these changes in a point-by-point reply to specific comments on the next pages.

**Line 27: Please add adequate sources to state that the primary resource gaps will grow. (Maybe, the ones in L69 will work?) L29: What do you mean by saying "the food portfolio [...] has been complicated by and increased degree of risks..."? L30: Please provide sources that the MENA region shows tendencies for an inability to satisfy needs with domestic production. L32: You say that (food) trade has been understudied - one might argue that as trade is a central part of food security (which you likewise support), it is quite well understood by the relevant trading actors. L29, 33: I think, MENA & VWT (and all other abbreviations) should be defined in the text (not in the abstract).**

➔ We applied reviewer's comments and revised the introduction by adding more references about the situation of the MENA region.

**Page 1: Line 27– Line 36**

Food security and water scarcity are urgent socio-economic and environmental issues in the Middle East and North Africa (MENA) region (Saladini et al., 2018), which are highly interlinked, and Water-Energy-Food Nexus has been suggested as a proper and integrated approach for resource management (Bazilian et al., 2011; Rasul, 2014; Mohtar and Daher, 2014; Lee et al., 2018). For example, food security in the MENA region has become complicated by increased risks owing to the geopolitical challenges and inability to satisfy needs with domestic production because of the lack of adequate arable land and water resources (Rastoin and Cheriet, 2010). In addition, food imbalance in the MENA region is forecast to reach 60 % in 2050 and food security in MENA region could be extremely compromised (Rastoin and Cheriet, 2010). Climate change could lead to more frequent occurrence of extreme climatic events in Mediterranean region, accompanying 50 % decrease of agricultural production by the end of the century (Porter et al., 2014). In particular, water saving through food trade can be suggested as a solution for mitigating groundwater depletion in the MENA region (Lezzaik et al., 2018).

**Concerning the meaning of VWT: if a product uses 1000 l/kg water to be produced in one region, it might have a much more severe impact in an arid climate than in a humid one (you cannot grow coffee in Lybia, but in Chile). If the value is to be interpreted locally, doesn't it lose its meaning and transferability?**

➔ We are not sure that we understood your comments correctly but we tried to answer your comments. We would like to explain the global water saving and national water saving by virtual water trade. If one country in arid region exports products to a country in humid region, global water saving would be negative value. But still the country in humid region could have water saving by importing products. However, some crops could limit to cultivate in some specific area, thus global water saving or national water saving in importing country was not meaningful but in exporting country water was used for producing exportable crops and it could convert to virtual water export.

**L56: You say that Fader et al (2011) show water savings of 263 km3/a due to beneficial agricultural production in other countries; does this calculation include the additional costs that arise from transport? Additionally, I am wondering how much the import of exotic products to western countries (an unnecessary trade in comparison to the import of basic crop products to arid countries) contributes to in the large savings (17 billion m3 blue water, L65) of global extent?**

➔ Water savings indicate the water requirement for producing the same amount of imported product, thus we hardly include additional cost for transportation. This study also did not consider the cost of transportation and energy parts, thus we added some paragraph about future works in conclusion.

**Page 10: Line 392 –Line 406**

However, this study only focused on food trade and water-land savings, thus energy part was not considered. The MENA region represents an extreme case globally in terms of water and energy resources, for example, 66% of the world's known crude oil reserves, but only 1.4% of the world's fresh water supplies is attributed to the region (Khater, 2001). The increase or decrease of water withdrawal for irrigation is related to the energy used for water extraction such as pumping surface or ground water. For example, 5 % or more of the total electricity consumption can be attributed to water pumping in Saudi Arabia (Siddiqi and Anadon, 2011). Energy use for food production and water supply could be the main factor in integrated resource management in the MENA region, and the lack of energy part was a limitation in this study.

In spite of this limitation, the intensity and connectivity of VWT, which were analyzed in this study, can be the major components needed for integrating resources management in the MENA region. Accordingly, VWT is regarded as the important factor in determining food security and water-lands management, and it can be a useful interlinking parameter among resources in WEF Nexus approach, which identify key issues in food, water, and energy securities through the lens of sustainability, seeking to predict and protect against future risks and resource insecurities (Biggs et al., 2015). The core of the Nexus concept is that the production, consumption, and distribution of water, energy, and food, are inextricably interlinked, thus this study would provide important information to policy makers for evaluating scenarios about integrated resource management toward sustainability in the MENA region.

**L111: please add units to WS/LS.**

➔ Yes, I added it.

**L114/115: Two sentences starting with "In addition" - please revise. I also do not understand the meaning of "In addition, each variable is dependent on local characteristics."**

➔ I thought these sentences were relevant, thus I removed them.

**L118: If you irrigate a crop with rain harvested water, either directly as water is used from the reservoir or indirectly as the reservoir water is used for enhanced groundwater recharge, is this blue or green water?**

➔ As followed by definition of green water by Falkenmark, it is the water captured by soil and used by crops. Thus, first we can calculate the soil moisture and crop water requirement, and if soil have enough water from rainfall for crop evapotranspiration, we do not need to irrigate. However, soil does not have enough water, we supply water by irrigation facility. But some irrigated water can go through ground water or runoff. Thus, technically speaking the green water indicate the amount of soil moisture which is used by evapotranspiration, and blue water indicated the amount of irrigation water used by evapotranspiration.

**L120: "Thus, the study for national water footprint should be executed for each country, basin, or specific area; however, this was outside the scope of the current study." -this sentence is unclear to me, especially the first part: what is the difference between "national" and "country"? For which regional unit did you carry out your study?**

➔ Mekonnen and Hoekstra (2010) estimated water footprint of each country in the world including the MENA region, thus water footprint applied in this study was country level data. We revised a little the paragraph about the water footprint reference.

**Page 3: Line 120 – Line 123**

Water footprint is a localized index for countries, accounting for the climate, productivity, and irrigation. In this study, we considered water footprints of all countries in the world, however, a lot of effort should be required for estimating water footprints of all countries and it was outside the scope of the current study. Therefore, we applied water footprint data of 147 countries, including those in the MENA region, from the study executed by Mekonnen and Hoekstra (2010).

**Can you please name the countries of the MENA region that you studied in the beginning, e.g. around L87ff?**

➔ We mentioned all name of countries of the MENA region that were considered as study countries.

**Page 3: Line 99 – Line 101**

The aim of this study is to evaluate the effects on water savings and land tenure from importing crops   at 15 countries in the MENA region such as Algeria, Egypt, Iraq, Jordan, Kuwait, Lebanon, Libya, Morocco, Oman, Qatar, Saudi Arabia, Syria, Tunisia, UAE, and Yemen.

**L127: What is the "limited water footprint"?**

➔ We removed it.

**Table 1: - Do I understand it correctly that the information in Table 1 is taken from Mekonnen and Hoekstra (2010)? If so, please add this information in the table caption. - Please add the time period of the data in the caption. - Can you please explain why the blue water footprint is larger than the green water footprint? Why does a plant need less rainwater than groundwater? - Which footprint did you use to calculate the land footprint?**

➔ We added caption in Table 1.

* Water footprint data was referenced by Mekonnen and Hoekstra (2010)

** Land footprint was calculated by crop production and cultivated area provided from World Bank open data (https://data.worldbank.org/)

➔ If there is not enough soil moisture from rainfall, irrigation should be required, thus if rainfall is very low, blue water requirement could be large than green water. Mekonnen and Hoekstra (2010) estimated the green and blue water footprint of various crop in more than 200 countries and reported them. More details about the calculation of green and blue water footprint is provided in https://waterfootprint.org/media/downloads/TheWaterFootprintAssessmentManual_2.pdf

**Table 2: - Again, please add the source of this data in the caption. - This data is shown for the years 2000-2012; I assume there are all mean values - please add this information. - If these are mean values, what was the standard deviation of the data? Is there a trend in the data? - Can you please add how this data was acquired and certain this data is? - Can you add a row showing the sums of the individual columns?**

➔ We revised the Table 2 with your comments.

**L154: It is good that you list previous network-based approaches that investigated VWT structures, but you should not only mention the citations and rather shortly summarize their works and how your work contributes to this.**

➔ We added summary of referenced studies.

**Page 5: Line 165 –Line 170**

A few studies have been conducted on the analysis of the structure of the VWT using a network-based approach (Konar et al., 2012; Dalin et al., 2012; Lee et al., 2016). For example, Konar et al (2012) analyzed the characteristics of the network change in virtual water trade (VWT), and found that a number of export trade partners followed an exponential distribution in 2000. Dalin et al (2012) found that constant organizational features were observed in the network of VWT even though the number of trade connections and the volume of VWT has been growing. In addition, Lee et al (2016) analyzed vulnerability of the importing countries through the characteristics of network in VWT.

**Equations 6 & 7: - is "j" in the sums as the starting counter equal to 1? I think, the usage of "j" is misleading, as it also refers to exporting countries. - is N (total number of countries) constant for all i (importing countries)? What if a country i only trades with one other country, i.e. N = 1; then, the equation gives a division by zero, correct? Equation 7: Why is the SInDC not related to the total volume of virtual water traded but to the number of total number of countries?**

➔ N is the number of entire network, thus it is constant to every country i, In addition, degree centrality is relative index for comparing country and N is constant number for all countries, thus the application of total number or total volume is not different for results.

**L172 & 173: I think, it should be "high levels" and "low levels".**

➔ We revised it.

**Eq 8: What is _alpha_ij?**

➔ We tried to clarity the methodology for Eigenvector centrality and added some example researches.
   **Page 5: Line 194 – Page 6: Line 216**
   In general, connections to nodes which are themselves influential could make a node more influence than connections to less influential nodes (Newman, 2016), and eigenvector centrality can be used for measuring the influential connections (Ruhnau, 2000). For example, the concept of eigenvector centrality has been used by the Web search engine Google in order to rank Web pages (Berry and Browne, 2005; Bryan and Leise, 2006; Newman, 2016).
   In VWT network, the eigenvector centrality could be used for identifying influential countries that could affect the entire network. In other words, the entire VWT can be affected by a few influential countries, and it is important to identify these countries for understanding and estimating the change of the entire structure of the VWT. An eigenvector centrality can measure the influence of each country in the entire VWT, and it is related not only to its own connection pattern but also to the connections of other countries to it. Therefore, a country is more influential if it is considered in relation to the countries that are influential themselves (Ruhnau, 2000). The eigenvector centrality assigns relative centrality to all of the countries in the VWT, based on the principle that connections to high-level centrality countries contribute more to the centrality of the countries compared to equal connections to low-level centrality countries (Ruhnau, 2000; Lee et al., 2016). Bonacich (1972) defined the centrality $(x_i)$ of a node i as the positive multiple of the sum of adjacent centralities in links (or volume) between nodes $(A_{ij})$. Therefore, if we denote the centrality of vertex i by $x_i$ , then we can allow for this effect by making $x_i$ proportional to the average of the centralities of i's network neighbours (Newman, 2016),
   $$x_i = \frac{1}{\lambda}\sum_{j=1}^{n} A_{ij}x_j \qquad (8)$$
   where λ is a constant. Defining the vector of centralities x = $(x_1, x_2,...)$, we can rewrite this equation in matrix form as
   $$\lambda x = Ax \qquad (9)$$
   This type of equation is solved using eigenvalues and eigenvectors, where A is a adjacency matrix of $A_{ij}$, and λ is a scalar, known as the eigenvalue associated with the eigenvector c defined as a column vector. Eigenvector centrality is determined by calculating the principal eigenvector that has the largest eigenvalue among all eigenvectors. A non-negative eigenvector with the maximal eigenvalue exists. We refer to a non-negative eigenvector $(x \geq 0)$ of the maximal eigenvalue as the principal eigenvector, and we call the entry $x_i$ the eigenvector-centrality of node (country) i (Ruhnau, 2000).

**L196ff - Please revise this paragraph: - The first sentence rather belongs to a summary, after you showed results, but you did not at this place in the manuscript. - The second sentence is given without reference/citation. - The third sentence contradicts the first two sentences. - The fourth sentence does not state whether Egypt imports from MENA countries or somewhere else. - The fifth sentence is not justified by the one example you state. - I also do not understand the intension of this paragraph, what do you want to convey here? Even the following sentence in L202 starts with "however" as if you wanted to say "but I actually want to talk about something else".**

➜ We revised the paragraph.

    **Page 6: Line 219 – Line 223**

    This study considered trade-offs between food security and food trade in terms of national resource management. For example, the increase of domestic food products instead of imports of them could be one policy for food security but additional water and land for domestic products would be considered at the same time. In other words, food imports could contribute domestic water and land management, therefore, we estimated the national water and land savings by importing crops as shown in Table 3.

**L206: "This means that the contribution of import of barley, maize, and wheat on water security in Saudi Arabia was significant." - how do you come to this conclusion?**

➜ We added internal water resource of each country in the MENA region into Table 1, which was provided from World Bank, and compared the amount of water saving with the internal water resource.

    **Page 6: Line 223- Line 228**

    In Saudi Arabia, blue water savings by barley, maize, and wheat imports were estimated to 5.0, 2.0 and 0.8billion m³/year, respectively. In comparison to the internal water resource of Saudi Arabia which is 2.4 billion m³/year as shown Table 1(World Bank, 2014), the water saving through import of barley, maize, and wheat could be considered as significant amount in Saudi Arabia. In the case of Egypt, most of the water saving occurred based on the imports of wheat and maize. Approximately 7.5 billion m³/year of blue water was saved by importing wheat. Specifically, the internal water resources in Egypt are only 1.8 billion m³/year (Table 1), therefore, water scarcity could be an issue for food security policy in Egypt.

**A general comment: for example, in L209, you state that Egypt would suffer from water shortage if the exporting countries banned wheat export to Egypt. I think that this is only partly true, i.e. only in those cases where the respective crops would actually grow in the individual countries. Considering rice, for example: I am sure that none of the MENA countries would be able to grow this crop even if the virtual water equivalent would be available. Please elaborate on this comment.**

➜ First, we need to explain the difference between water saving and virtual water import.

    Virtual water import was based on water use in exporting country, thus virtual water import by rice could be quantified in terms of exporting country even through rice could not be suitable for growing in the MENA region.

    Water saving is kinds hypothetical number in this study because we assumed that all products were produced in domestically, thus we did not include rice in water saving part. However, the results of water saving could bring the importance of food import and showed how much water would be required for domestic production.

**Page 4: Line 133– Line 144**

Food import is also related to domestic water and lands savings. In particular water saving has a different meaning from virtual water import. For example, Saudi Arabia imported wheat from various exporters and virtual water import indicates the sum of the products obtained from multiplying the quantity of imported wheat by the respective water footprint of each exporter. However, water saving indicates the amount of water needed to produce the same quantity of imported products domestically. Therefore, water saving by wheat import in Saudi Arabia is estimated by multiplying the quantity of imported wheat with the water footprint of wheat in Saudi Arabia.

In this study, we applied green and blue water footprints of crops in each country in the MENA region, as shown in Table 1. However, the availability of water footprint data in the MENA region was limited in some cases. For example, the water footprint of wheat was available in all countries except for Bahrain. Lands saving has the same implication as water savings, thus we calculated lands saving using land footprint of each country in the MENA region, as shown in Table 2. The land footprint indicates the land requirement for producing 1 ton of crops, and it was calculated based on the harvest area and crop production data collected from FAOSTAT (Table 1).

**L208: The statement of 1.8 billion m3/a water available for Egypt is missing a source.**
➔ We added source of internal water resource in Table 1

**L210: "The crop import could result in a large amount of land savings." - this is an unnecessary statement. Likewise in L215: "These results can elicit useful information for analyzing the trade-off between food and water-land securities in the MENA region in terms of sustainable development."**
➔ We removed those expressions and revised whole paragraph.

**L210ff: "In Saudi Arabia, land savings based on the import of barley, maize, and wheat, amounted to 1.6 million ha/year, and Lebanon was also strongly influenced by the impact of crop import on land savings. For example, approximately 0.24 million ha could be saved by crop imports, comprising 36% of the agricultural area in Lebanon, that indicates that the crop trade in Lebanon has significant benefits in terms of land resources compared to water resources." - please revise and do not mix two different countries in different sentences.**
➔ We revised those sentences.

**Page 6: Line 228 – Line 231**

Lebanon was strongly influenced by the impact of crop import on land savings. Approximately 0.24 million ha could be saved by crop imports, comprising 36% of the agricultural area in Lebanon, that indicates that the crop trade in Lebanon has significant benefits in terms of land resources compared to water resources.

**L216: What do you mean by this: "However, water saving indicates the virtual water saving, and sometimes it is larger than the total water resources in some countries. "**
**L216/217: "However" twice as starting word.**
**L217: "However, these results showed that the increase of food security is accompanied by**

numerous water requirements in the MENA region." - I do not understand this. Please revise. L218ff: "Additionally, the saved land is not always suitable for agricultural areas." – The "saved land", i.e. the equivalent required area to grow imported crops, is probably not available. Do you have information on this? "Some crops are required for the specific type of land, ..." - It is rather the other way: you require a specific soil for this or that crop."...and the productivity is also different based on soil." - Do you mean "the productivity is varies with different soils"? "Even if we can save land..." - Why do you think, the reason to import is to save land? - Why do you write "we"? "...there is the limitation for considering the land saving as an agricultural land saving in accordance to this study." - What do you mean by this?

➔ We thought that above all comments were related to the same paragraph, and soil part was not related to this paper. Thus, we revised them. In revised paragraph we meant the limitation of virtual water trade, and removed the soil part.

**Page 6: Line 232– Line 238**

Food imports could be regarded as a negative factor in food security, and it is obvious that food security would accompany water and lands for domestic food products. These results showed that food imports could bring positive impacts on numerous water and lands savings in the MENA region. However, there are limitations of these results. First, water saving estimated in this study was based on the hypothetical situation that meat there were no international trade situation, and sometimes it was larger than the internal water resources in some countries such as Saudi Arabia and Egypt. Additionally, some crops are required for the specific type of climate but this study assumed that MENA region was suitable for cultivating maize, wheat, barley, and rice.

**Table 3: - Please check for unnecessary line breaks (eg. Saudi Arabia, Blue water, Barley). - Do I understand it correctly that table 3 shows the results from the product of water footprint (table 1) and the annual import (table 2)? If so, how could you fill the gaps for the water footprint in blue water barley and green water maize? -> Oh, I see you wrote "0" for partly - please correct this and write "-".**

➔ We revised Table 3.

**Section 3.1 should be shortened; often, statements are given that are unnecessary, unproven or uncited. The information from table 3 can and should be offered in a much more compact way.**

➔ We revised the entire section 3.1.

**L227: Are the numbers for annual water import average values?**

➔ Yes, it is average value, thus we mentioned the "average" in revised manuscript.

**Fig 1: - The grey scale (ie the total water import) uses uneven separating numbers and unequal intervals; I suggest to use even numbers (e.g. 1500 - 15000 instead of 1495 -15410 for the first green water import interval) and evenly spaced intervals. - I cannot read the number in the legend for annual water import - Some pie charts are very small (Qatar, Oman, Bahrain, Lebanon) - Why do the pie charts vary in size?**

➔ We removed the pie chart and focused on total virtual water import from 2000 to 2012.

**Table 3 vs 4: I do not understand the difference between "water savings due to imported crops" (table 3) and "imported water" (table 4) - can you please explain this difference and describe why both values are different?**

➔ We added more explanation about the differences between water saving and virtual water import.
**Page 4: Line 133– Line 144**

Food import is also related to domestic water and lands savings. In particular water saving has a different meaning from virtual water import. For example, Saudi Arabia imported wheat from various exporters and virtual water import indicates the sum of the products obtained from multiplying the quantity of imported wheat by the respective water footprint of each exporter. However, water saving indicates the amount of water needed to produce the same quantity of imported products domestically. Therefore, water saving by wheat import in Saudi Arabia is estimated by multiplying the quantity of imported wheat with the water footprint of wheat in Saudi Arabia.

In this study, we applied green and blue water footprints of crops in each country in the MENA region, as shown in Table 1. However, the availability of water footprint data in the MENA region was limited in some cases. For example, the water footprint of wheat was available in all countries except for Bahrain. Lands saving has the same implication as water savings, thus we calculated lands saving using land footprint of each country in the MENA region, as shown in Table 2. The land footprint indicates the land requirement for producing 1 ton of crops, and it was calculated based on the harvest area and crop production data collected from FAOSTAT (Table 1).

**Section 3.2.2 / figure 3: how could you determine which water (blue or green) was used to grow the crops in the exporting countries?**

➔ Mekonnen and Hoekstra (2010) estimated green and blue water footprint of each country in the world including the MENA region, thus we used green and blue water footprints applied in this study was country level data. We revised a little the paragraph about the water footprint from their study.

**Fig. 3: Why do you give the numbers here in Gm3 while all other volumes are given as volume / time (Mm3/y)? I suggest to be consistent for comparability especially with such large numbers which are hard to imagine.**

➔ We changed the unit to Mm3/yr.

**Fig. 5: This is a very nice interpretation, but I have a suggestion: you could combine a and b and connect the individual countries' marks with arrows; currently, one has to search for a long time before a country's performance can be compared.**

➔ We changed the order of Figures, thus previous Fig.5 is Fig. 6 in the revised manuscript.
We added the arrows in Fig. 5.

**Fig. 6: - Please check for non-discribed countries and/or add them to "others". – The numbers of the individual eigenvectors are too small and cannot be read. - Can you show this figure also for the whole MENA region? Or in other words: why did you choose Lebanon here? Is the figure similar for the other countries?**

➔ We changed the order of Figures, thus previous Fig.6 is Fig. 4 in the revised manuscript.
We made a new figure of the MENA region, and others indicate the countries who export less than 100 Mm³/yr to the MENA region or Lebanon

**L359: If you write "Since the introduction of the virtual water concept, various studies have been conducted to quantify the volume of the VWT." you should provide proper citations and describe how you contribute to an extension of their findings.**

➔ We thought this sentence is already mentioned in Introduction, thus we removed it in Conclusions.

**L361: As above, the statement "The amount of imported virtual water is regarded as the most important factor in determining water and food security," should be backed up by citations or proof.**

➔ Actually, that statement was derived from the results from this study, thus we revised them in Conclusions.

**L364: "...the interlinkages of key natural resource sectors and the improved production efficiency are considered a win–win strategy for environmental sustainability..." - I do not understand why you address production efficiency here; that was not part of you previous analysis. Can you please explain this?**

➔ We agreed with your opinion, thus removed that sentence.

**L368: "Thus, decisions made in one sector typically impact the other sectors." - I think that this statement here does not belong to your core message of the paper: you never discuss / analyze how different sectors influence each other. You also do not show how virtual water or changes in virtual water fluxes may influence whatever sector.**

➔ We agreed with your opinion, thus removed that sentence.

**L372: "...policy makers can benefit..." - how should they benefit? What would be the key parameter policy makers can use? How should they decide on the future if your study is only based on the analysis of data from the past? Also: you compared the different countries of the MENA region among each other and derived values for SInDC and NSInDC. The comparison is thus only a qualitative comparison. How should a single country decide now whether its food import strategy generally is stable?**
**Finally: considering political differences in the MENA region, do you think that any singular country or a coalition of countries could use your evaluation to increase its food stability?**

➔ Still, it is limitation of virtual water concept that it is hard to apply virtual water to real policy. We tried to study some real cases, but it is still lack of the study. We keep trying to find the appropriate example.

➔ We added more sentences about the contribution of this study in terms of policy making.

Page 10: Line 381– Line 391

The import of water in virtual form based on VWT could develop into a major water portfolio that dominates water management in the water-scarce countries of the MENA region. In water-deficit areas, such as the MENA region, the VWT can offer new perspectives for understanding and solving water stress and scarcity. In summary, this study showed that the significant water in comparison to internal water resource could be saved by food trade in the MENA region, and policy makers can benefit by considering both the quantitative impacts of VWT and the structural changes of VWT, such as vulnerable expansion (or reduction) in the MENA region. For example, when a country in the MENA region set a plan for increasing food security, this country first should identify the amount of water and land savings that can be achieved by food import, and consider the trade-off between food security and food import. In addition, the stable trade could be a component for stable food supply in the MENA region, thus this study contributes to the understanding of the dependency on each trade partner for countries in the MENA region and can help with setting the food trade policy in terms of extension (or reduction) of trade partners and increase (or decrease) in volume of trade.

**Page 10: Line 399– Line 406**

In spite of this limitation, the intensity and connectivity of VWT, which were analyzed in this study, can be the major components needed for integrating resources management in the MENA region. Accordingly, VWT is regarded as the important factor in determining food security and water-lands management, and it can be a useful interlinking parameter among resources in WEF Nexus approach, which identify key issues in food, water, and energy securities through the lens of sustainability, seeking to predict and protect against future risks and resource insecurities (Biggs et al., 2015). The core of the Nexus concept is that the production, consumption, and distribution of water, energy, and food, are inextricably interlinked, thus this study would provide important information to policy makers for evaluating scenarios about integrated resource management toward sustainability in the MENA region.

**Point-by-point reply to specific comments**
**Referees #2**

**Reviewer's Comment: The water footprints of a given crop vary widely by country: for barley, green WF ranges from 193.6 to 6417.6 m3/ton. Adding together green and blue still gives a very wide range: ～8200 m3/ton in Libya vs 1000 m3/ton in Saudi Arabia. Are these numbers and their spatial variability realistic? Is it possible that producing barley in Libya consumes 8 times as much water as in Saudi Arabia? I don't imagine that potential ET varies that much over the region. Is the very wide range in WF because yields are so much higher in Saudi Arabia, but water consumption is assumed to be independent of yield? Some explanation is needed.**

**Answer:** In this study, national water footprint of various crops from Mekonnen and Hoekstra, 2010 was applied. In my opinion, water footprint is affected by not only crop water requirement but also productivity. Thus, even if there is not much big difference in crop water requirement based on ETc, the productivity at each country in MENA region could be huge different. For example, the production and cultivated area of barley in Libya provided from World Bank were 191,641 ha and 94,107 ton, thus the productivity is 0.49 ton/ha but Saudi Arabia has 5.67 ton/ha (12,279 ha, and 68,366 ton). It was almost 10 times difference. Therefore, the difference of productivity could be one of main reason of wide range of water footprint.
* * *
**Reviewer's Comment: I found the methods description for Eigenvector centralities confusing (L176-193). Please rewrite for clarity.**

**Answer:** We tried to clarity the methodology for Eigenvector centrality and added some example researches.

**Revision: Page 5: Line 194 – Page 6: Line 216**
In general, connections to nodes which are themselves influential could make a node more influence than connections to less influential nodes (Newman, 2016), and eigenvector centrality can be used for measuring the influential connections (Ruhnau, 2000). For example, the concept of eigenvector centrality has been used by the Web search engine Google in order to rank Web pages (Berry and Browne, 2005; Bryan and Leise, 2006; Newman, 2016).
In VWT network, the eigenvector centrality could be used for identifying influential countries that could affect the entire network. In other words, the entire VWT can be affected by a few influential countries, and it is important to identify these countries for understanding and estimating the change of the entire structure of the VWT. An eigenvector centrality can measure the influence of each country in the entire VWT, and it is related not only to its own connection pattern but also to the connections of other countries to it. Therefore, a country is more influential if it is considered in relation to the countries that are influential themselves (Ruhnau, 2000). The eigenvector centrality assigns relative centrality to all of the countries in the VWT, based on the principle that connections to high-level centrality countries contribute more to the centrality of the countries compared to equal connections to low-level centrality countries (Ruhnau, 2000; Lee et al., 2016). Bonacich (1972) defined the centrality $(x_i)$ of a node $i$ as the positive multiple of the sum of adjacent centralities in links (or volume) between nodes $(A_{ij})$.

Therefore, if we denote the centrality of vertex i by $x_i$ , then we can allow for this effect by making $x_i$ proportional to the average of the centralities of i's network neighbours (Newman, 2016),

$$x_i = \frac{1}{\lambda}\sum_{j=1}^{n} A_{ij}x_j \tag{8}$$

where λ is a constant. Defining the vector of centralities x = $(x_1, x_2,...)$, we can rewrite this equation in matrix form as

$$\lambda x = Ax \tag{9}$$

This type of equation is solved using eigenvalues and eigenvectors, where A is a adjacency matrix of $A_{ij}$, and λ is a scalar, known as the eigenvalue associated with the eigenvector c defined as a column vector. Eigenvector centrality is determined by calculating the principal eigenvector that has the largest eigenvalue among all eigenvectors. A non-negative eigenvector with the maximal eigenvalue exists. We refer to a non-negative eigenvector $(x \geq 0)$ of the maximal eigenvalue as the principal eigenvector, and we call the entry $x_i$ the eigenvector-centrality of node (country) i (Ruhnau, 2000).
* * *
**Reviewer's Comment: Some numbers are claimed to be significant, but without context. For example, Saudi Arabia saves 2 billion m3 per year by importing barley. Is that big number? Compared to what?**

**Answer:** In previous manuscript, it was difficult to evaluate the results of water savings. Therefore, we added internal water resource of each country in MENA region into the Table 1, and compared the water savings with the internal water resource.

**Revision: Page 6: Line 223 – Line 229**
**Answer:** In Saudi Arabia, blue water savings by barley, maize, and wheat imports were estimated to 5.0, 2.0 and 0.8billion m³/year, respectively. In comparison to the internal water resource of Saudi Arabia which is 2.4 billion m³/year as shown Table 1(World Bank, 2014), the water saving through import of barley, maize, and wheat could be considered as significant amount in Saudi Arabia. In the case of Egypt, most of the water saving occurred based on the imports of wheat and maize. Approximately 7.5 billion m³/year of blue water was saved by importing wheat. Specifically, the internal water resources in Egypt are only 1.8 billion m³/year (Table 1), therefore, water scarcity could be an issue for food security policy in Egypt. Lebanon was strongly influenced by the impact of crop import on land savings.
* * *
**Reviewer's Comment: The authors correctly note that it is important to identify countries that rely on only a few exporters. I'm not so sure that this means that countries with high dependence on one exporter should re-evaluate their policy, since I don't know enough about international trade strategy. Is there literature that can show that, historically, countries that rely on a single exporter are vulnerable to food sanctions? Can the authors cite historical precedent? Also, I found the shift in exporting countries from the US and Australia to other nations of potential importance to explain, both its causes and consequences. Is there more you can say about that in the paper?**

**Answer:** We tried to search for historical precedent about impacts of trade structures on the international trade strategy. However, we could not find the specific examples. In terms of geopolitical issues and historical data use, we added some paragraph as limitations and future work parts.

**Revision: Page 9: Line 358 – Line 363**

[revised manuscript text omitted]
 for domestic products should would be considered at the same time required for increasing domestic production. In other words, food imports could contribute domestic water and land management, Ttherefore, we estimated we estimated the national water and land savings by importing crops as shown in Table 3., that is a negative factor for food security. In Saudi Arabia, Table 3 shows that the green and blue water savings by barley, maize, and wheat imports in Saudi Arabia were estimated to 5.0, 2.0 and 0.82.0 and 7.8 billion m³/year, respectively. In comparison to the internal water resource of Saudi Arabia which is 2.4 billion m³/year as shown Table 1(World Bank, 2014), This means that the water saving through contribution of import of barley, maize, and wheat could be considered as significant amount on water security in Saudi Arabia. was significant. In the case of Egypt, most of the water saving occurred based on the imports of wheat and maize. Approximately 7.5 billion m³/year of blue water was saved by importing wheat. Specifically, the internal water resources in Egypt are only 1.8 billion m³/year (Table 1), therefore, water scarcity could be an issue for food security policy in Egypt. Lebanon was also strongly influenced by the impact of crop import on land savings. For example, aApproximately 0.24 million ha could be saved by crop imports, comprising 36% of the agricultural area in Lebanon, that indicates that the crop trade in Lebanon has significant benefits in terms of land resources compared to water resources.

Food imports could be regarded as a negative factor in food security, and it is obvious that food security would accompany water and lands for domestic food products. These results showed that food imports could bring positive impacts on numerous water and lands savings in the MENA region. These results can elicit useful information for analyzing the trade-off between food and water land securities in the MENA region in terms of sustainable development. However, However, there are limitations of these results. First, wwater saving estimated in this study wasis based on the hypothetical situation that meatns there wereare no international trades situation, indicates the virtual water saving, thus and and sometimes it wasis larger than the total 
[revised manuscript text omitted]